# MEMORIES TO MAPS: MECHANISMS OF IN-CONTEXT REINFORCEMENT LEARNING IN TRANSFORMERS

## ABSTRACT

Humans and animals show remarkable learning efficiency, adapting to new environments with minimal experience. This capability is not well captured by standard reinforcement learning algorithms that rely on incremental value updates. Rapid adaptation likely depends on episodic memory—the ability to retrieve specific past experiences to guide decisions in novel contexts. Transformers provide a useful setting for studying these questions because of their ability to learn rapidly in-context and because their key-value architecture resembles episodic memory systems in the brain. We train a transformer to in-context reinforcement learn in a distribution of planning tasks inspired by rodent behavior. We then characterize the learning algorithms that emerge in the model. We first find that representation learning is supported by in-context structure learning and cross-context alignment, where representations are aligned across environments with different sensory stimuli. We next demonstrate that the reinforcement learning strategies developed by the model are not interpretable as standard model-free or model-based planning. Instead, we show that in-context reinforcement learning is supported by caching intermediate computations within the model's memory tokens, which are then accessed at decision time. Overall, we find that memory may serve as a computational resource, storing both raw experience and cached computations to support flexible behavior. Furthermore, the representations developed in the model resemble computations associated with the hippocampal-entorhinal system in the brain, suggesting that our findings may be relevant for natural cognition. Taken together, our work offers a mechanistic hypothesis for the rapid adaptation that underlies in-context learning in artificial and natural settings.

## 1 INTRODUCTION

Animals can learn efficiently and rapidly adapt to new environments with minimal experience. For example, humans can infer underlying structure or learn new concepts from just a handful of examples and mice in maze tasks can identify optimal paths after only a few successful trials (Meister, 2022). Standard reinforcement learning (RL) algorithms, which typically rely on incremental value updates to shape decision-making, does not capture this rapid learning behavior well (Eckstein et al., 2024). One explanation is that biological agents possess useful priors shaped by evolution and experience, allowing them to generalize quickly in naturalistic settings. They also rely on episodic memory, the ability to recall specific past experiences to guide decisions in novel situations.

Here, we ask how episodic memory operates not just as storage, but as a computational substrate for rapid learning and decision-making. We train a transformer model to perform in-context reinforcement learning (Lee et al., 2023) on navigation tasks inspired by rodent behavior. In each new environment, the model receives exploratory trajectories as context and infers a goal-directed policy. Transformers are especially relevant not only because of their established capabilities for rapid and flexible in-context learning (Dong et al., 2022; Brown et al., 2020; Lampinen et al., 2024), but also because their key–value memory architecture has been linked to models of episodic memory in the brain (Krotov & Hopfield, 2020; Tyulmankov et al., 2021; Fang et al., 2025). Understanding the reinforcement learning strategy that emerges in these models can provide new hypotheses for how memory-based computations might support flexible decision-making in new environments.

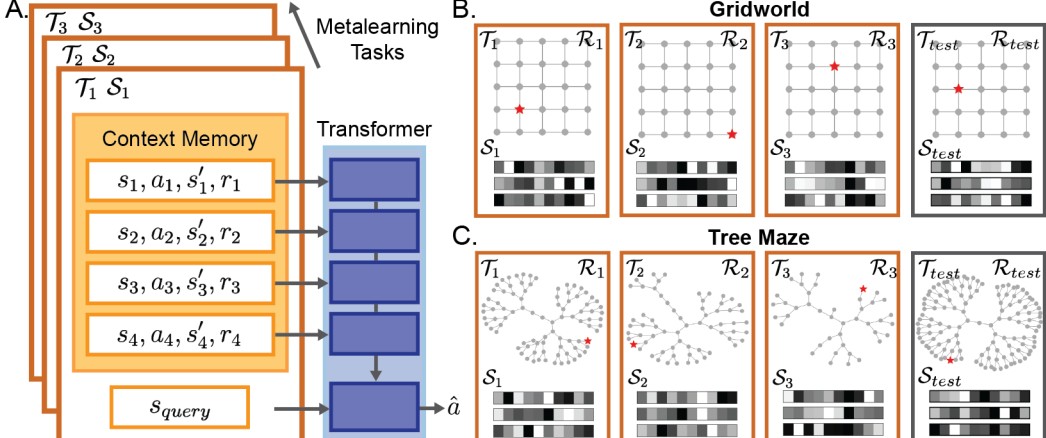

Figure 1: **A transformer is trained to in-context reinforcement learn in diverse planning tasks.**
**A.** Diagram of meta-learning setup. For each task, the model is trained via supervision to predict
the optimal action from a query state $s_{\text{query}}$, given memories of RL transition tuples sampled in-
context. **B.** Illustration of three training tasks (orange) and one test task (gray) from the gridworld
distribution. In each task, the underlying graph structure is fixed, but the reward location (red star)
can vary. Each state is encoded as a random Gaussian vector (bottom). Importantly, test task state
encodings are novel. **C.** As in (B), but for the tree maze distribution. The training set graph structures
are drawn from probabilistically branching trees, while the test set structure is a full binary tree.

We focus on two task suites: spatially regular gridworlds and hierarchically structured tree mazes
(Fig 1BC). While both require memory to support goal-directed behavior, they differ sharply in ge-
ometry: gridworlds are Euclidean and spatially continuous, whereas tree mazes are non-Euclidean
and branch-structured. This contrast allows us to evaluate how learned in-context strategies gen-
eralize across structural regimes known to challenge standard sequence models– for instance, lan-
guage models are known to struggle with symbolic reasoning and hierarchical generalization in
tree-structured domains (Bogin & Berant, 2022; Ruiz & Nachum, 2021; Keysers et al., 2020).

In this paper, we make the following contributions:

- We show that transformers trained to in-context reinforcement learn develop consistent
  representation learning strategies: structure learning within contexts, and alignment of rep-
  resentations across contexts with shared regularities.

- We demonstrate that the model learns computations found in natural cognition. Representa-
  tion learning strategies are consistent with suggested roles for hippocampus and entorhinal
  cortex, and memory recall patterns at decision time are consistent with hippocampal replay.

- We give descriptions for the in-context RL algorithms that emerge. We show with mecha-
  nistic analysis that the model does not use standard model-free or model-based RL meth-
  ods. Instead, strategies tend to rely on intermediate computations cached in memory tokens,
  demonstrating how episodic memory can be used as a computational workspace.

**Related works** In meta-RL settings, the outer learning loop shapes the weights of the network to
learn an algorithm that can be deployed in-context. The in-context learning within each task occurs
via activation dynamics– through memory and internal state– rather than parameter updates (Beck
et al., 2023; Sandbrink & Summerfield, 2024; Lampinen et al., 2024). Early examples of this ap-
proach include RL², which meta-trains a recurrent neural network using RL in the outer loop, such
that an in-context RL strategy emerges in the inner loop (Wang et al., 2016; Duan et al., 2016). Sub-
sequent work extended this approach to include explicit episodic memory mechanisms, combining
RNNs with key–value memory architectures (Ritter et al., 2018, 2020; Team et al., 2023). More
recently, Lee et al. (2023) proposed decision-pretrained transformers (DPTs), which use supervised
training in the outer loop to induce in-context RL behavior in the inner loop. We adopt DPTs in
this work both for practical reasons–their scalability and ease of training–and for scientific ones:

the transformer architecture allows us to probe how memory-based computation supports in-context learning. This latter motivation is inspired by recent findings suggest that key–value architectures, like the transformer, offer a useful computational analogy for episodic memory (Krotov & Hopfield, 2020; Ramsauer et al., 2020; Tyulmankov et al., 2021; Whittington et al., 2021; Fang et al., 2025; Chandra et al., 2025). See App. A for a more thorough discussion of related work.

## 2 EXPERIMENTAL METHODS

**Meta-learning procedure**  We adopt the decision-pretraining framework from Lee et al. (2023) (Fig. 1A), training models via supervision to learn optimal policies from in-context experience. Each task is a Markov Decision Process defined by state encoding function $\mathcal{S}$, action space $\mathcal{A}$, transition function $\mathcal{T}$, reward function $\mathcal{R}$. For each task, the model receives an in- context dataset $\mathcal{D}$ of RL transition tuples $(s, a, s', r)$ gathered from an exploratory policy, plus a query state $s_{\text{query}} \in \mathcal{S}$. The model is meta-trained to predict the action from an oracle policy. At test time, the model generalizes to held-out tasks with novel sensory observations using only in-context information, demonstrating in-context reinforcement learning. We focus on offline settings where $\mathcal{D}$ comes from random exploration, and more details can be found in App. B.

**Structure of task suites**  Our first task distribution is a $5 \times 5$ gridworld in which the reward location is fixed but hidden from the agent. This setting is loosely inspired by the Morris water maze, a behavioral task used to study how animals use memory to navigate unknown environments (Vorhees & Williams, 2006). Across tasks, $\mathcal{T}$ is fixed, while $\mathcal{S}$ and $\mathcal{R}$ vary (Fig. 1B). At test time, the model is deployed in a gridworld with novel sensory observations and a new reward location. Our second task distribution consists of tree-structured mazes, which introduce hierarchical state transitions and sparse rewards. These are settings where rodents have been shown to display rapid learning (Rosenberg et al., 2021). The meta-training set consists of binary trees generated with some branching probability so that $\mathcal{T}$ varies across tasks (Fig. 1C). $\mathcal{S}$ and $\mathcal{R}$ also vary across tasks. The action space consists of four options: stay, move to the parent node, or move to either child node. At test time, the model is evaluated on a full 7-layer binary tree, consistent with Rosenberg et al. (2021), again using novel state encodings not seen during training.

In both tasks, states are represented by 10-dimensional random vectors. Full task details are in App. B. Together, these two tasks allow us to analyze model behavior in spatially regular environments and branching tree structures (which may be a relevant analogy for language generation tasks).

**Model architecture and selection**  Our base architecture is a causal, GPT2-style transformer with 3 layers and 512-dimensional embeddings. We provide the context memory $\mathcal{D}$ before the query token $s_{query}$. This ordering supports an interpretation in which previous experiences are stored as cached key–value memories that are retrieved at query, or decision-making, time. Additional details on architecture and training are in App. C. We also test alternative modeling choices in App. D.

## 3 RESULTS

### 3.1 TRANSFORMERS LEARN A RL STRATEGY TO RAPIDLY SOLVE PLANNING TASKS.

We first evaluate the agent's performance in new environments with novel sensory observations.

**Gridworld**  In held-out gridworld environments, we test the agent with query states that were observed in-context and located at least 6 steps from the goal. We plot return as a function of context length (Fig. 2A; App. E). The meta-learned agent often navigates directly to the reward after a single exposure, mirroring one-shot learning reported in rodents navigating water mazes (Steele & Morris, 1999). To summarize test environment performance, we plot return as a function of the number of rewards experienced in-context (Fig. 2B, blue). As expected from Fig. 2A, the agent achieves near-maximal performance after just one exposure, with only minor improvements thereafter.

We next compare the meta-learned model to standard reinforcement learning methods. Specifically, we train a tabular Q-learning agent and a deep Q-network (DQN) on each test environment. Each Q-learning agent is trained using a replay buffer containing the same in-context dataset $\mathcal{D}$ provided to the meta-learned model (App. F). We again summarize performance in Fig. 2B. The performance gap between both Q-learning agents and the meta-learned model is substantial but expected, reflect-

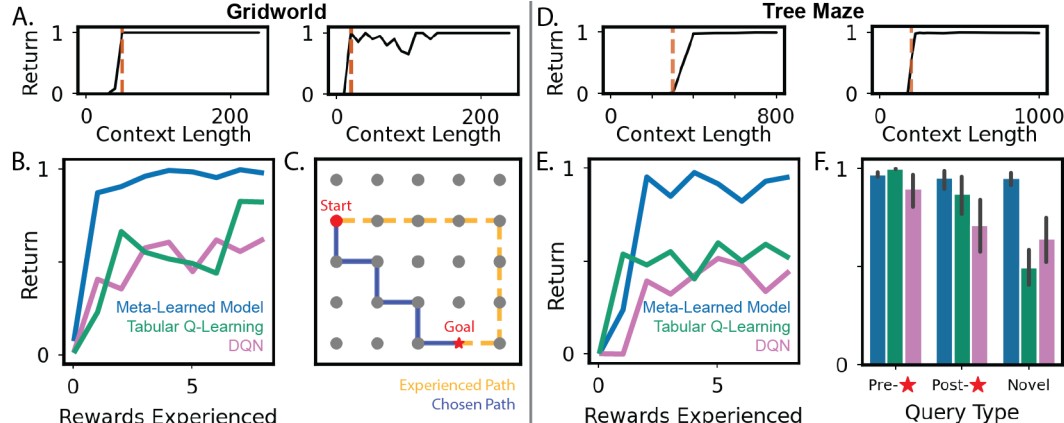

Figure 2: **Transformers can rapidly learn and plan in new tasks. A.** Average max-normalized return in two held-out gridworld environments as a function of context length. For each context length, 20 query states are sampled with test horizon 15. **B.** As in (A), but return is plotted against the number of rewards experienced in-context and averaged over 50 held-out environments. Blue: meta-learned transformer; Green: tabular Q-learning; Pink: DQN. **C.** Example of shortcut behavior in a held-out gridworld. The model experiences a circuitous trajectory (orange), but can infer a more efficient path (blue). **D., E.** As in (A, B), but for tree mazes and test horizon 100. **F.** As in (D,E), but shown only for context length 800 and subdivided by query type: states seen before reward (Pre-⋆), after reward (Post-⋆), or never seen in context (Novel; not used in E). Error bars show 95% C.I.

ing the utility of meta-learned priors. The advantage of the tabular agent over DQN demonstrates that representation learning adds additional difficulty in novel environments. Overall, describing the learning efficiency seen in animals may require moving beyond single-task RL frameworks.

Finally, we observe that the meta-learned agent discovers shortcut paths to reward. Even when the agent only observes a circuitous path in-context, it infers a policy that selects the shortest route to reward—often through previously unseen states (Fig. 2C). Quantitatively, the model selects shortcut paths in over 60% of test simulations (App. E). This suggests that the agent has internalized the Euclidean geometry of the environment, a feature that we analyze more deeply in later sections.

**Tree Mazes** We next evaluate the meta-learned agent in test tree mazes, where the agent rapidly learns the task after only a few reward exposures (Fig. 2DE; App. E). As before, the meta-learned agent captures rapid learning more effectively than Q-learning baselines (Fig. 2E). The tabular agent again outperforms the DQN, confirming that representation learning remains a core challenge.

To gain insight into the priors acquired through meta-learning, we evaluate all models at a long context length (800 timesteps). We stratify performance by the type of query state (Fig. 2F). When the query state had already been seen prior to any reward, the tabular agent performed comparably to the meta-learned model. However, when the query state was seen only after the final reward, the tabular agent underperforms. In standard Q-learning, states encountered only along paths away from reward do not receive value propagation. This suggests that the meta-learned model acquires a useful prior: the ability to infer inverse actions. Finally, we evaluate performance when the query state was never encountered during context. In this setting, the meta-learned agent performs better than both Q-learning baselines, likely due to a learned prior over action selection.

### 3.2 RL STRATEGY SHAPES REPRESENTATIONS VIA IN-CONTEXT STRUCTURE LEARNING.

The behavioral results in Fig. 2B,E highlight that representation learning poses a key computational challenge in these tasks. How does the model organize state representations—and does a structured representation learning strategy emerge during in-context processing?

**Gridworld** We begin by visualizing low-dimensional embeddings of model activity across gridworld states. Because the query token represents the agent's current state, we extract its activity as the basis for representation analysis. We project the 512-dimensional query representations into a 2D

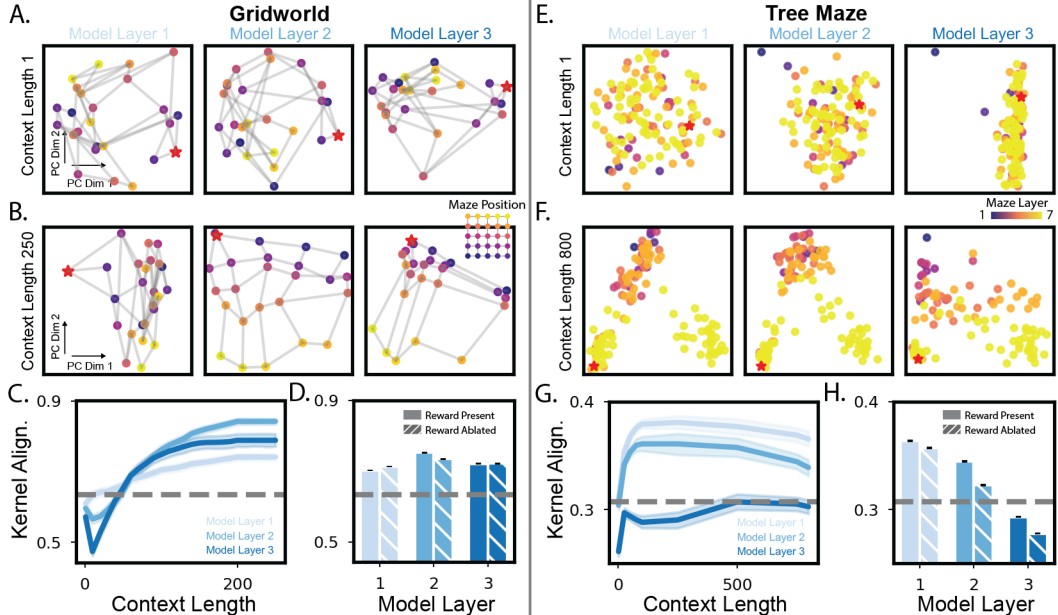

Figure 3: **Model representations are shaped by in-context structure learning. A.** An example test gridworld. Query token representations are visualized for each state after projection onto the first two principal components, for layer 1 (left), layer 2 (middle), and layer 3 (right). Context length is 1. Points are colored by graph location; gray lines indicate true connectivity. Reward is marked with a red star. **B.** As in (A), but with context length 250. **C.** Kernel alignment between model representations and latent graph structure as a function of context length, across 100 environments. Shading shows 95% C.I.; colors denote model layer. Dashed line shows baseline from raw inputs. **D.** As in (C), but for context length 250, with reward ablation (shaded bars). **E, F.** As in (A, B), but for test tree mazes. Points are colored by maze depth. **G, H.** As in (C, D), but for test tree mazes.

PCA space at each model layer, for both short and long context lengths (Fig. 3A–B; App. G). With limited context, the model's representations are disorganized and show no spatial structure (Fig. 3A). As in-context experience increases, the representations become more structured and reflect the latent geometry of the gridworld (Fig. 3B). This structure resembles predictive representation learning, but crucially, no such objective was imposed during training.

To quantify this, we compute the kernel alignment between model representations and the latent environment structure across held-out tasks (Fig. 3C; App. H). Kernel alignment increases with context length, with layer 2 consistently exhibiting the strongest correspondence to latent structure. Surprisingly, representation structure is largely unaffected by the presence of reward (Fig. 3D; App. G, H). Overall, we find that in-context experience induces geometry-aligned state representations.

**Tree Mazes** Does in-context representation learning also emerge in agents trained on the tree maze task? We repeat the PCA projection analysis on query token representations in tree mazes (Fig. 3E–F; App. G). As context increases, representations organize into a bifurcating structure that roughly mirrors the maze's hierarchical layout (Fig. 3F), which suggests that the model learns coarse, high-level structure rather than fine-grained spatial layout. Consistent with this intuition, kernel alignment also increases with context length in tree mazes (Fig. 3G), but remains lower than in the gridworld task. Representations in the tree maze are more strongly modulated by reward (Fig. 3H; App. G, H).

Taken together, these results indicate that the model meta-learns in-context representation strategies that vary in granularity across task domains. This provides **normative support for the hypothesis that structure learning facilitates efficient decision-making**, and in fact in-context structure learning has also been found in the representations of large language models (Park et al., 2024). This observation parallels predictive map formation in the hippocampus (Stachenfeld et al., 2017)—long hypothesized as a computational scaffold for memory—and suggests that similar principles can emerge in artificial agents through meta-learning.

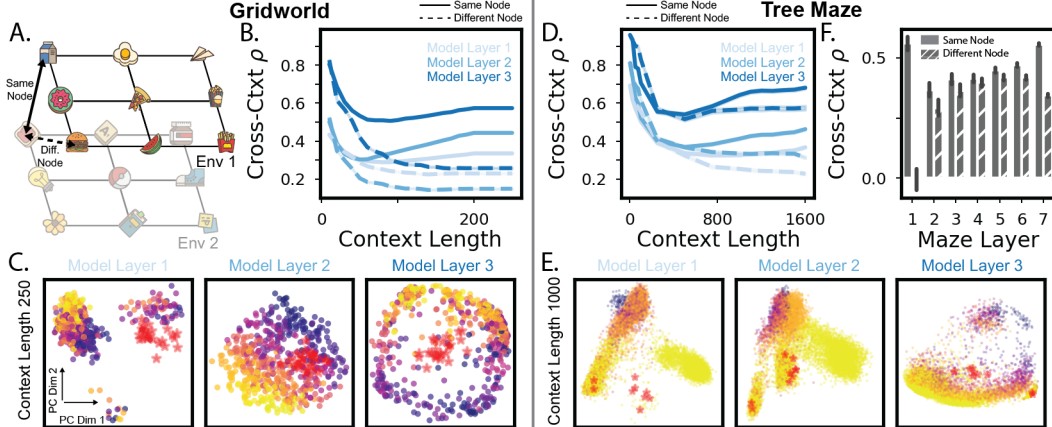

Figure 4: **As context grows, representations across environments with similar structure are aligned. A.** Diagram of cross-environment alignment (Whittington et al., 2020). Although sensory inputs differ, environments share latent structure, and representations of matching latent states should be similar. **B.** Average pairwise Pearson correlation coefficient of node representations across 100 gridworld environments, as a function of context length. Solid lines: same-node comparisons. Dashed lines: different-node comparisons. Shading shows 95% C.I.. Line color denotes model layer. **C.** PCA visualization of representations pooled from 15 randomly selected visualizations. **D.** As in (B), but for 50 tree mazes. **E.** As in (C), but for tree mazes. **F.** Summary of (D) at context length 1600, averaged across layers. X-axis denotes the maze-layer of the comparison node.

### 3.3 REPRESENTATIONS ARE REUSED ACROSS ENVIRONMENTS WITH SHARED STRUCTURE.

In neuroscience, the hippocampal–entorhinal circuit is thought to support structure learning across contexts (Buckmaster et al., 2004; Kumaran et al., 2009; Whittington et al., 2020). A leading hypothesis holds that the hippocampus encodes context-specific experiences, while the entorhinal cortex abstracts shared structure across environments (Whittington et al., 2020). Do similar cross-context alignment strategies emerge in meta-learned agents?

**Gridworld**    A key signature of cross-context structure learning is the alignment of internal representations across environments with shared topology: even when sensory observations differ across environments, states occupying the same grid location should be encoded more similarly than states from different locations (Fig. 4A). To test this, we compute pairwise correlations between model representations of the same graph node across different test environments (Fig. 4B). We separate correlation scores by whether the compared states occupy the same or different graph nodes. With limited context, representations appear collapsed—showing high correlation across all states regardless of node identity. As context length increases, this gap widens: states from the same node become more aligned than those from different nodes (Fig. 4B; solid vs. dashed).

This alignment is also visually apparent when directly inspecting the model's internal representations. We aggregate representations from 50 test environments and project them into a shared 2D PCA space (Fig. 4C). In layers 2 and 3, representations from corresponding graph nodes cluster across environments, reflecting shared latent structure. Notably, representations of goal states also align across environments, despite the reward location being randomized.

**Tree Mazes**    We repeat the same analyses in tree mazes and find similar alignment strategies emerge (Fig. 4DE). We further analyze the correlation scores by the node position within the tree (Fig. 4F). Cross-context alignment is strongest for states near the root or leaves of the tree. This is consistent with §3.2, where the representations capture coarse structure over precise positional detail in tree mazes. Importantly, neither in-context nor cross-context representation alignment was explicitly trained—these strategies emerge opportunistically as a byproduct of meta-learning.

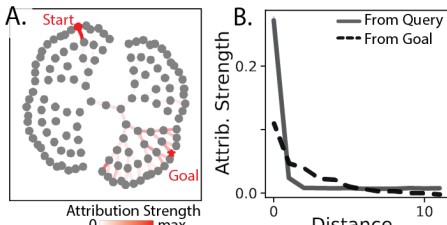

Figure 5: **Memory retrieval at decision time shows limited expansion from the query state and the goal state. A.** Example tree maze environment, context length 800. Edge color indicates gradient attribution strength for each transition. **B.** Average attribution strength of each context memory vs distance from query state and goal state, across 50 environments.

### 3.4 RL STRATEGY IS NEITHER VALUE-BASED LEARNING NOR MODEL-BASED PLANNING.

Thus far, we have shown that meta-learned agents replicate the rapid learning dynamics observed in animal behavior. A key component of this success is the emergence of structured representations from contextual input. We now turn to characterizing the mechanisms of the underlying RL strategy. We begin by testing whether the model exhibits hallmarks of standard model-free reinforcement learning. We test whether value information can be linearly decoded from model representations, but overall did not find evidence for this (Apps. K and L).

Next, we test whether the model exhibits hallmarks of standard model-based RL, which typically requires path planning from query to goal. Planning need not follow a strictly forward rollout, and transformers in particular can implement diverse state-tracking strategies (Li et al., 2025). Critically, however, all such strategies depend on retrieving intermediate states along the path from query to goal during decision time. To evaluate this, we assess which context-memory tokens influence the model's decision at a given query state. Using integrated gradients, we measure attribution strength for memory tokens along the query→goal path (Fig. 5; App. M). In both tasks, only tokens near the query and goal states show high attribution. As a further test, we also conduct attention ablations and find similar results (App. M). Both results are inconsistent with path planning, which requires attending to transitions along the full route at decision time.

In summary, we find that the agent does not use value gradients or path planning to make decisions. We suggest that the learned in-context strategy lies outside the standard taxonomy of model-free and model-based reinforcement learning. Our analyses also reveal an additional neural prediction: memory retrieval at decision time should be biased toward experiences near the agent's current location and its goal. Such replay patterns have been observed in the hippocampus during spatial decision-making tasks (Jackson et al., 2006; Pfeiffer & Foster, 2013; Mattar & Daw, 2018).

### 3.5 MODELS LEARN STRATEGY WHERE INTERMEDIATE COMPUTATIONS ARE STORED IN CONTEXT-MEMORY TOKENS.

We now aim to describe the algorithms used by the model to plan in each task. To do so, we first review the roles of query and memory tokens. The query token encodes the agent's current state, while memory tokens represent previously observed transitions. During inference, the query token attends to memory tokens to integrate past experience into its policy computation. Across layers, both query and memory tokens are updated with newly computed features, allowing memory to serve as an active computational substrate. To reveal how computation unfolds, we will focus on understanding which tokens are critical in each layer and what information is contained in tokens.

**Gridworld** In gridworld tasks, we suggest that the following strategy is used by the model:

1. Use in-context experience to align representations to Euclidean space.
2. Given a query state, calculate the angle in Euclidean space between query and goal state.
3. Use the calculated angle to select an action in that direction.

We arrived at this hypothesis by first identifying the task-relevant variables that can be linearly decoded from the query token at each layer (Fig. 6A). We train a linear decoder to predict the underlying XY position of the query state from its embedding (App. K) and evaluate accuracy on held-out environments with novel sensory observations. Decoding accuracy improves across layers, with spatial position becoming reliably recoverable by layer 2 (error $< 1$; Fig. 6B). Building on this spatial structure, we next decode the angle from the query state to the goal– again, this information can be accurately decoded by layer 2 (Fig. 6C). Interestingly, both XY position and angle-to-goal can be decoded from the embeddings of the context-memory transitions $(s, a, s', r)$ as well.

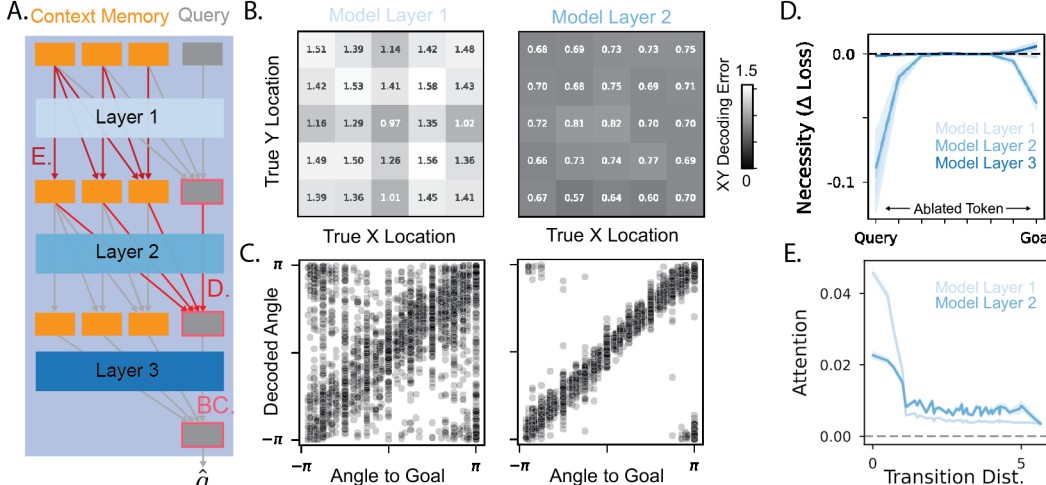

Figure 6: **Gridworld tasks are solved by aligning internal representations to Euclidean space.**
**A.** Overview of analysis steps. **B.** Total XY decoding error across 60 test environments, plotted
by true XY position of the query state, given query token embeddings from layer 1 (left) and layer
(right). Context length = 250. **C.** As in (B), but for decoded vs. true angle from query state to
the goal. **D.** Change in cross-entropy loss after ablating context tokens along the query–goal path,
plotted by ablated token position. Line color indicates layer of intervention. **E.** Average attention
score between context-memory tokens as a function of spatial distance between tokens, for layers 1
and 2. (D,E) show mean over 50 environments, with shading for 95% C.I.

To localize where angle-to-goal information may be computed, we test which context-memory to-
kens are necessary for correct decisions. Using attention ablations, we show that model performance
relies on attending to tokens near the query and goal states in layer 2 (Fig. 6D, App. N). We pro-
pose that layer 2 extracts the internal XY coordinates from query and goal state tokens to compute
the relative angle between them. To understand how XY information arises, we examine how state
representations evolve through context-to-context attention.

We show that, across layers, attention patterns between context memory tokens shift from localized
to distributed (Fig. 6E, App. N), suggesting that the model first stitches transitions locally before
constructing global structure. Overall, we suggest that the model organizes memory to reflect Eu-
clidean structure and use that geometry to guide action selection. This explains the model's ability
to take unseen shortcuts (Fig. 2C).

**Tree Mazes**   In tree mazes, a useful strategy can be to identify when the agent is on a critical path
to reward and to default to the parent-node action otherwise. This is because there are only 6 states
in the maze (of 127) where the optimal action is to transition to the left child or right child (Fig 7B).
These are the states on the path from root to reward, which we will call the left-right (L-R) path.
Indeed, the model has a strong action bias to take parent-node transitions (App. O). Overall, we find
evidence that the model exploits this structure and learns the following strategy:

1. Use in-context experience to stitch transitions backwards from the goal to root, and tag
   context-memory tokens that are on the L-R path.
2. Given a query state, check if there are context-memory tokens that contain the query state
   and are on the L-R path. If not, default to taking the parent-node transition.
3. Otherwise, extract the optimal action information from the tagged context-memory tokens.

We arrive at this hypothesis by asking how the model takes correct actions when it is on the L-R path.
Again, we work backwards and analyze which context-memory tokens are sufficient to influence the
output from the final model layer (Fig 7C, App. O). Surprisingly, we find that the model output is
unaffected if the query token of the last layer attends only to context-memory tokens involving the
query state. These tokens contain sufficient information for the model to make its decision.

With this in mind, our next question was to understand what information is contained in the context-
memory tokens entering the last model layer. We repeat our linear decoding analyses on the context-

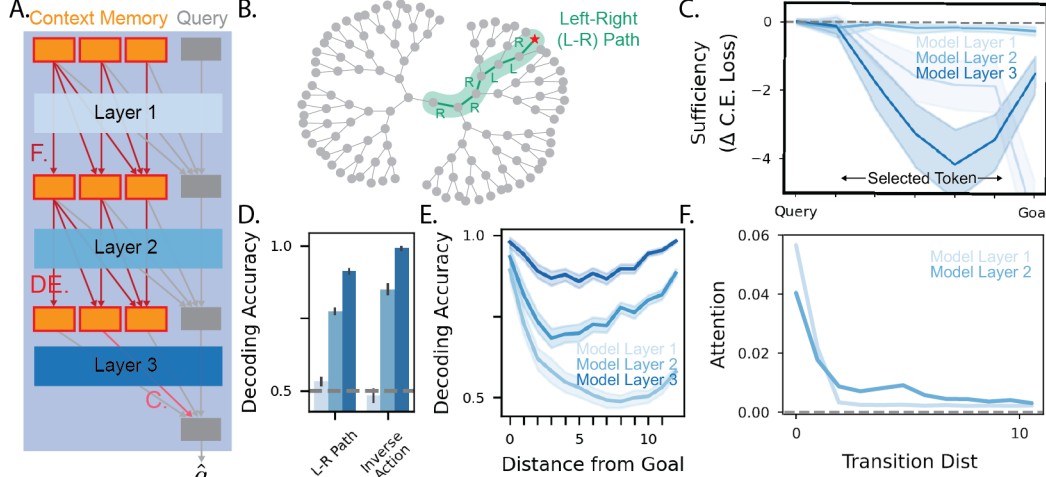

Figure 7: **Tree mazes are solved by tagging context-memory tokens on a critical path to reward.**
**A.** Overview of analysis steps. **B.** We focus on the path from root to goal (L-R path). **C.** Change in cross-entropy loss when attention is fixed to context-memory tokens at specific points on the L-R path. In all tests, the query state is the root. Line color indicates layer of intervention. Mean across 50 environments, shading shows 95% C.I. **D.** Given embeddings of context-memory transitions, the balanced accuracy of decoding their presence on the L-R path and the inverse action. Decoding the inverse action is only non-trivial for parent-node transitions so we test only on those. Embeddings are taken from the input tokens to each model layer (line color). Mean across 60 environments, error bars show 95% C.I. Dashed line is chance. **E.** L-R Path decoding accuracy from (D), but separated by how far each context-memory transition is from goal. **F.** As in Fig 6E but for tree mazes.

memory tokens. Two variables are well-decoded. First, the inverse action for the transition represented in a context-memory token can be decoded with high accuracy (Fig 7D). The other well-decoded variable is whether the context-memory token is a transition on the L-R path (Fig 7D), regardless of direction (i.e., towards or away from goal). Possibly, at decision time the model tests if there are context-memory tokens that contain the query state and are tagged as being on the L-R path. If so, then the correct left/right action can be inferred from the same tagged tokens (in particular since inverse actions are also encoded). We find further evidence for this strategy by re-doing our sufficiency analysis from Fig 7C with additional restrictions on the selected tokens (see App. O).

Finally, we ask how this information becomes present in the context-memory tokens. We plot the L-R path decoding accuracy from Fig 7D by the distance from the context-memory token to goal (Fig 7E). Across model layers, we see that the accuracy first improves for tokens closest to or farthest from the goal. Later, the accuracy improves for tokens at an intermediate distance from the goal, where more transition information must be integrated to know if the token is on the L-R path. Furthermore, attention patterns between context memory tokens shift from localized to distributed across model layers (Fig 7F). Taken together, we suggest that path stitching occurs between context-memory tokens such that L-R path tokens are tagged expanding backwards from the goal.

## 4 CONCLUSION

We have shown that rapid adaptation of agents in tasks relevant to natural cognition can be explained by RL strategies that lie outside traditional model-free or model-based frameworks. Despite this, our meta-learned model also displays phenomena expected from neural activity: learning of environment structure, alignment of representations across environments, and biased memory recall patterns at decision-time. Taken together, this suggests that understanding the cognitive processes that support rapid learning may require theorists to consider a broader space of planning strategies.

Finally, our analysis of the RL strategies that emerge in transformers suggest a novel use of episodic memory– each memory is not only a record of the original experience, but also stores additional computation useful for decision making (Dasgupta & Gershman, 2021).

## REPRODUCIBILITY

Code will be publicly available in a Github link in the final paper (after de-anonymization).

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
