# A RELATED WORKS

**Meta-learning to discover in-context reinforcement learning algorithms** In order to develop a model that can in-context reinforcement learn, we use a recently introduced meta-learning framework (Lee et al., 2023). Meta-learning is concerned with "learning-to-learn", using prior knowledge learned from previous tasks to support rapid adaptation to new ones (Beck et al., 2023). That is, the goal is to learn an algorithm $f$ that can be deployed in new tasks. The learning of $f$ is called the *outer-loop* while $f$ itself is referred to as the *inner-loop* (Beck et al., 2023). In many settings where $f$ is a RL algorithm (meta-RL), the outer loop shapes the weights of the network, but learning within each task occurs via activation dynamics– through memory and internal state– rather than parameter updates (Beck et al., 2023; Lampinen et al., 2024). The resultant $f$ is considered an in-context learning algorithm (Beck et al., 2023; Sandbrink & Summerfield, 2024; Lampinen et al., 2024).

Early examples of this approach include $RL^2$, which meta-trains a recurrent neural network using reinforcement learning in the outer loop, such that an in-context RL strategy emerges in the inner loop (Wang et al., 2016; Duan et al., 2016). Subsequent work extended this approach to include explicit episodic memory mechanisms, combining RNNs with key–value memory architectures (Ritter et al., 2018; 2020; Team et al., 2023). More recently, Lee et al. (2023) proposed decision-pretrained transformers (DPTs), which use supervised training in the outer loop to induce in-context RL behavior in the inner loop. We adopt DPTs in this work both for practical reasons–their scalability and ease of training–and for scientific ones: the transformer architecture allows us to probe how memory-based computation supports in-context learning.

**Meta-learning to describe cognition and neural activity** Meta-learning has been proposed as a framework to model both cognitive flexibility and structured learning in neuroscience and psychology (Binz et al., 2024). In human cognitive tasks, meta-learned models have been used to replicate observed heuristics in decision-making and to account for few-shot generalization (Dasgupta et al., 2020; Binz et al., 2022; Lake & Baroni, 2023). Meta-RL, in particular, has been used to generate hypotheses about how neural systems implement learning across tasks. Previous studies have used the RL2 framework to show how the outer and inner learning loops can model different areas of the brain, with the prefrontal cortex often playing a key role (Wang et al., 2018; Hattori et al., 2023; Zheng et al., 2025). Despite these advances, many computational models in neuroscience rely on single-task RL, training agents independently on each task without leveraging prior experience. In contrast, we use meta-RL as a tool for developing flexible in-context learning, without attempting to localize the outer loop to any specific brain region. Our focus is on the computational content of the learned representations and decision-making strategies.

**Transformers and episodic memory systems** Transformer models process sequences by computing self-attention over key–value pairs, enabling flexible access to information across long contexts (Vaswani et al., 2017). This key–value structure has led to interpretations of transformers as memory systems (Geva et al., 2020), aligning them with a broader class of models that incorporate explicit memory mechanisms (Graves et al., 2014; Sukhbaatar et al., 2015; Graves et al., 2016; Banino et al., 2020). These systems separate memory addressing (keys) from memory content (values), enabling high-fidelity storage and targeted retrieval. This architectural separation bears many similarities to theoretical accounts of episodic memory in the brain (Teyler & DiScenna, 1986; Teyler & Rudy, 2007; Gershman et al., 2025). For example, recent work has formalized connections between key–value architectures and Hopfield networks, a classic model of associative memory in the brain (Krotov & Hopfield, 2020; Ramsauer et al., 2020). Related approaches such as fast-weight models (**??**) offer alternative mechanisms for temporary memory storage and in-context computation, often drawing from Hebbian or synaptic dynamics. Other studies have proposed biologically grounded implementations of key–value attention mechanisms, further linking transformer-like architectures to neural computation (Bricken & Pehlevan, 2021; Tyulmankov et al., 2021; Whittington et al., 2021; Kozachkov et al., 2023; Fang et al., 2025; Chandra et al., 2025). Several of these models take direct inspiration from the hippocampus, a brain region widely implicated in episodic memory (Whittington et al., 2021; Fang et al., 2025; Chandra et al., 2025). Recent experimental work has also identified key–value–like coding patterns in hippocampal activity during episodic memory tasks (Chettih et al., 2024). Together, these findings suggest that key–value architectures offer a useful computational analogy for episodic memory. In this work, we analyze a meta-trained transformer to examine what kinds of memory-supported strategies emerge when such an architecture is optimized for rapid adaptation.

## B TASK CONSTRUCTION

### B.1 GRIDWORLD

We use a $5 \times 5$ 2D gridworld environment. Thus, there are $N = 5 \times 5 = 25$ states in the environment, each of which corresponds to an underlying $(x, y)$ location. Actions are one-hot encoded and consist of: up, right, down, left, stay. If the agent chooses to take an action that hits the environment boundaries, this manifests as a "stay" transition. The transition structure in this environment $\mathcal{T}$ (that is, how actions transition the agent from one $(x, y)$ state to another) is fixed across all tasks. Each task is defined by the sensory encoding $\mathcal{S}$, the reward location $\mathcal{R}$, and the in-context exploration trajectory $\mathcal{D}$.

Each state in a gridworld task corresponds is encoded by a 10-dimensional vector. For each task, we describe the set of these $N$ encoding vectors as $\mathcal{S}$. The following describes how we generate the encoding vectors comprising $\mathcal{S}$. We first define a random expansion matrix $E \in \mathcal{R}^{N \times N}$, where $E_{i,j} \in \mathcal{N}(0, 1)$. We next construct a distance correlation matrix $D \in \mathcal{R}^{N \times N}$ by exponentiating the negative Euclidean distances between all pairs of grid positions: $D_{i,j} = \sigma^{(||(x_i,y_i)-(x_j,y_j)||_2)}$ for states $i$ and $j$ and their corresponding $(x, y)$ locations. Here, $\sigma \in [0, 1]$ is a correlation parameter that controls how strongly nearby positions are correlated in the encoding space. Thus, the encoding of state $i$ is computed as $\frac{ED_{:,i}}{||ED_{:,i}||_2}$ and $\mathcal{S} = \{ \frac{ED_{:,i}}{||ED_{:,i}||_2} \}_{i=0}^{N}$.

The reward state $\mathcal{R}$ is chosen from the $N$ states in the environment. The in-context exploration trajectory is generated from a random walk with a randomly chosen initial state, plus some reasonable heuristics. Specifically, we make the probability of selecting the "stay" action half as likely as the other actions. In addition, if the agent takes an action that causes it to not transition to a new state, the probability of taking that action again is downweighted to $0$ until the agent transitions to a new state (preventing the agent from getting stuck at boundaries). Running this biased random walk for $T$ steps gives us $C = (s_t, a, s'_t, r_t)_{t=0}^{T}$, a set of standard RL transitions.

Our dataset is generated offline before training. We now describe how we construct the train/evaluation/test sets. For a desired dataset size of $M$, we partition the $N$ states of the gridworld environment into three sets of sizes $M * p_{train}, M * p_{eval}, M * p_{test}$. Specifically, we divide the dataset with ratios: $p_{train} = 0.8, p_{eval} = 0.1, p_{test} = 0.1$. To generate the training dataset, we construct $M * p_{train}$ tasks, where we sample $\mathcal{R}$ from the corresponding training partition of states. We then sample $\mathcal{S}$ and $\mathcal{D}$ as described above. We repeat this for the evaluation and test datasets. The training dataset is used for pretraining. The evaluation dataset is used for validation during pretraining and selecting models. The test dataset is used for any analyses conducted after model training and selection.

### B.2 TREE MAZES

We use binary tree environments with 7 tree layers (that is, a minimum of 6 actions is needed to navigate from root to leaf). Actions are one-hot encoded and consist of: right child, left child, parent, stay. If the agent tries to transition to a node that does not exist (e.g. trying to go to "parent" from the root), this manifests as a "stay" transition. The transition structure $\mathcal{T}$ can vary across tasks. This is because in each task the underlying tree is generated with branching probability $0.9$. Thus, there is a maximum of 127 states in each task. Each task is defined by the transition structure $\mathcal{T}$, sensory encoding $\mathcal{S}$, the reward location $\mathcal{R}$, and the in-context exploration dataset $\mathcal{D}$.

The sensory encodings are generated as in gridworld, except $D$ is defined via the geodesic distances between any two tree states. The reward state $\mathcal{R}$ is chosen only from leaf nodes. The in-context exploration trajectory $\mathcal{D}$ is generated from a random walk from the root node, with reasonable heuristics. We use the same heuristics as in gridworld. We also add heuristics described in Rosenberg et al. (2021) of mice in similar mazes. That is, the agent is more likely to alternate between left and right transitions when transitioning through child nodes. In addition, the agent is less likely to backtrack.

As before, our dataset is generated offline before training. For a desired dataset size of $M$, we construct three sets of sizes $M * p_{train}, M * p_{eval}, M * p_{test}$, with $p_{train} = 0.8, p_{eval} = 0.1, p_{test} = 0.1$. To generate the training dataset, we construct $M * p_{train}$ tasks, where we sample $\mathcal{T}$

from binary trees with branching probability 0.9. We use only trees with at least one leaf node in the seventh layer and, for the training dataset, exclude the full 7-layer tree. We sample $\mathcal{R}$ from one of the leaf nodes. We then sample $\mathcal{S}$ and $\mathcal{D}$ as described above. We repeat this for the evaluation dataset, ensuring distinct $\mathcal{T}$ from the training dataset. The test dataset comprises only of $\mathcal{T}$ corresponding to a full binary tree, with $\mathcal{R}, \mathcal{S}, \mathcal{S}$ sampled as above. As b before, the training dataset is used for pretraining. The evaluation dataset is used for validation during pretraining and selecting models. The test dataset is used for any analyses conducted after model training and selection.

## C MODEL AND TRAINING PARAMETERS

We largely follow the same architecture as that of Lee et al. (2023), a GPT-2 style model with causal attention and without positional embeddings. Our default model has 4 heads, 3 layers, and embedding dimension of 512. Context memory tokens consist of the $(s_t, a, s'_t, r_t)$ tuple concatenated together into one vector. Thus, tokens are 26-dimensional in gridworld and 25-dimensional in tree maze. The query token consists of $(s_q, \vec{0})$ for query state $s_q$, where $\vec{0}$ provides 0-padding to reach the desired vector size. These tokens are projected into model embedding space through a learnable linear layer. The model samples greedily in the gridworld environment and with softmax sampling in the tree maze environment (both settings were empirically determined).

In contrast to Lee et al. (2023), we provide the query token at the end of the context memory. This is to allow a clearer interpretation in which context memory tokens represent previous experiences in the maze that are stored in episodic memory. The query token is only provided at decision time, and the agent must use previous memories to guide its present decision.

To allow for query tokens at the end of an input sequence sequence and to preserve efficient pretraining, we make modifications to the pretraining procedure of Lee et al. (2023), which we describe here. Let's say we have a pretraining task with context memory tokens $\mathcal{D}$ and query state $s_q$. To encourage length generalization, we would like to train the model on many sequence lengths– let's say every $t_{step}$ timesteps of $\mathcal{D}$. To do so from one forward pass, we first construct a sequence $\mathcal{D}_{train}$ where $s_q$ is interleaved every $t_{step}$ timesteps of $\mathcal{D}$: $\mathcal{D}_{train} = [\mathcal{D}_1, \mathcal{D}_2, \ldots, \mathcal{D}_{t_{step}}, s_q, \mathcal{D}_{t_{step}+1}, \ldots \mathcal{D}_{2*t_{step}}, s_q, \mathcal{D}_{2*t_{step}+1}, \ldots, \mathcal{D}_T, s_q]$. We then construct an attention mask $A_{mask} = A_{causal} + A_{query}$, where $A_{causal}$ is the standard causal attention mask with $-\infty$ values in the upper-triangular and 0 elsewhere. $A_{query}$ ensures that query tokens are not processed by context memory tokens by masking columns corresponding to the query token:

$$A_{query}[i,j] = \begin{cases} -\infty, & \text{if } \mathcal{D}_{train}[j] = s_q \text{ and } i \neq j \\ 0, & \text{otherwise} \end{cases} \tag{1}$$

Thus, we use $A_{mask}$ during training and cross-entropy loss is only calculated over the outputs corresponding to $s_q$. In gridworld, the maximum context length in training is $T = 200$. In tree maze, the maximum context length in training is $T = 800$.

We use Adam optimizer with weight decay $1 \times 10^{-5}$. We use a batch size of 1024 for gridworld and a batch size of 512 for tree mazes. We train the model for 25 epochs in gridworld and 50 epochs in tree mazes. We use a learning rate of $1 \times 10^{-4}$ which we linearly decrease to $1 \times 10^{-5}$ over the course of training (we found that this empirically worked well). For each training run, we use two NVIDIA H100 GPUs. This results in around 1 hour of training time for gridworld and 1.5 hours of training time for tree mazes. For each task, we train models from 5 random initializations, for both a dropout of 0 and 0.2. We then select the best model via validation performance for each task. This selected model is the model we analyze for each task.

We did not find much improvement by trying other tweaks to the pretraining setup or model size. We show validation and training results for various different parameters in figure 8 and 9.

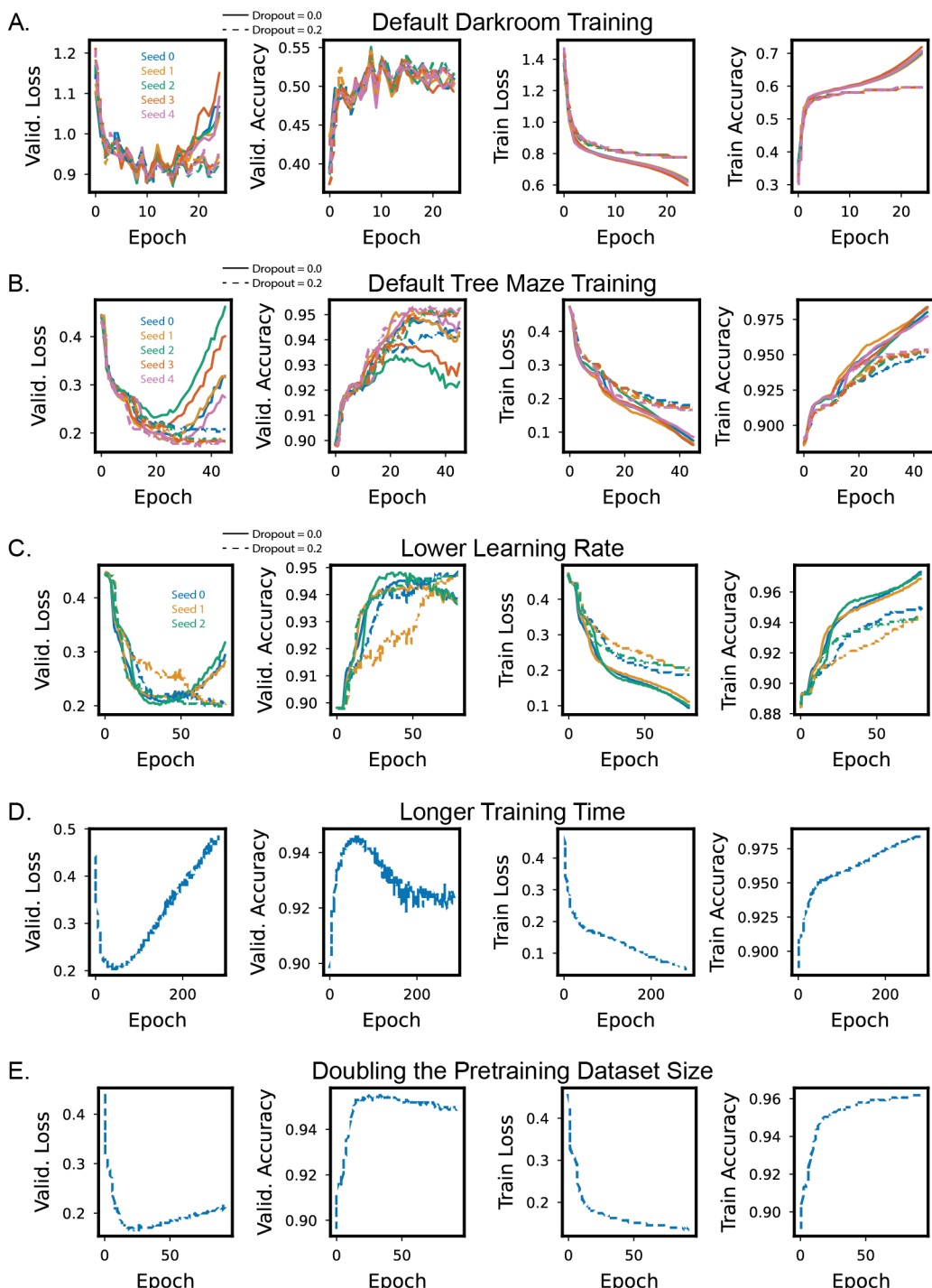

Figure 8: Effect of training parameters. **A.** Default training settings for Darkroom task, showing validation loss, validation accuracy, training loss, and training accuracy over training epochs. Colors indicate random seeds and line style indicates dropout amount. We note that $100\%$ accuracy is not possible due to the training procedure (see task construction details). **B.** As in (A), but for tree mazes. **C.** As in (B), but for $\frac{1}{10}$ of the default learning rate. We don't use a learning rate scheduler here. **D.** Seed 0 of (B), but we let the training run for 250 epochs. **E.** As in (B) but pretraining dataset is doubled in size.

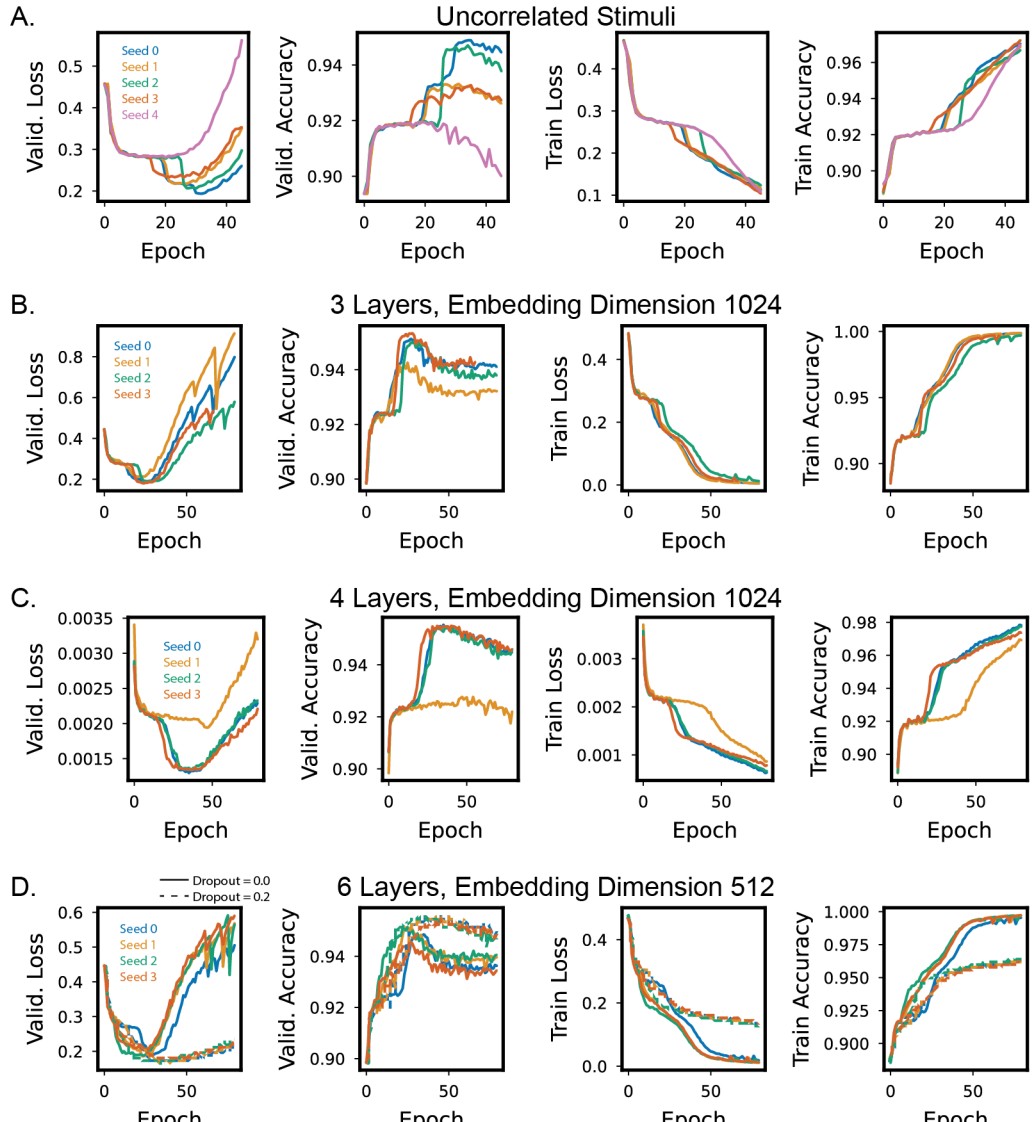

Figure 9: Effect of model and dataset parameters on training. **A.** As in Fig 8B, but state encoding is fully uncorrelated. **B.** As in Fig 8B, but model encoding dimension is doubled to 1024. **C.** As in Fig 8B, but model encoding dimension is doubled to 1024, and number of model layers is increased to 4. **D.** As in Fig 8B, but number of model layers is doubled to 6.

# D  RESULTS SENSITIVITY TO TASK/MODEL PARAMETERS

We repeat some of the analyses in the main figures here for alternative parameterizations of task and model. We focus mostly on testing the tree maze environment, for simplicity.

We first test the model where stimuli do not have spatial correlation (Fig 10). We find similar coarse in-context representation structure emerges, where representations from the first layer roughly separate out the two main branches of the maze (Fig 10AB). However, the bifurcating structure is less clear than it is when some spatial correlation is introduced (Fig 3 and Fig16). The cross-context structure results seem similar to that of the correlated stimuli (Fig 10C, compare to (Fig 4F). The bias of attribution scores to the query and goal at decision time also mirrors that seen in environments with correlated stimuli (Fig 10DE, compare to (Fig 5BC).

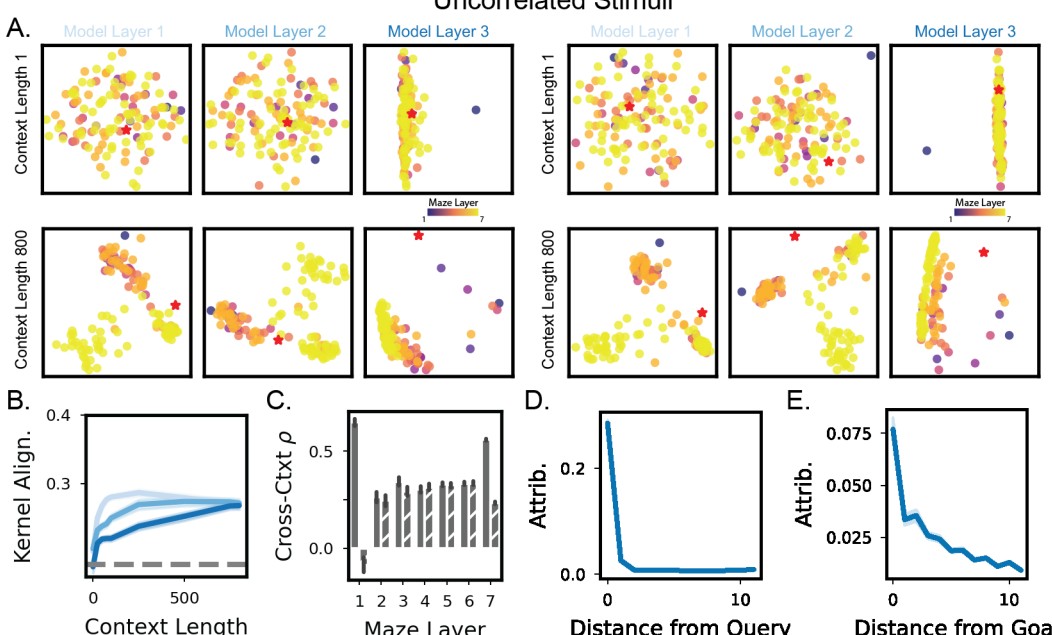

Figure 10: Sensitivity of results to state encoding correlation. **A.** As in Fig 3E, but for two random example environments (left and right). Additionally, the model was pretrained and tested on environments with uncorrelated stimuli. **B.** As in Fig 3C, but for uncorrelated stimuli. **C.** As in Fig 4F, but for uncorrelated stimuli. **D., E** As in Fig 5BC, but for uncorrelated stimuli.

We next test a larger version of the model with 6 layers (Fig 11). Like before, the in-context representation structure emerges as a bifurcating structure in the middle layers of the model (Fig 11AB). With more layers, though, the representations of the first layer now appear disorganized. As before, the last layer of the model is also organized in a less interpretable structure. The cross-context structure results again reflect greater latent structure alignment in the early and late layers of the model (Fig 11C, compare to (Fig 4F). The bias of attribution scores to the query and goal at decision time also mirrors that seen in environments with correlated stimuli (Fig 11DE, compare to (Fig 5BC).

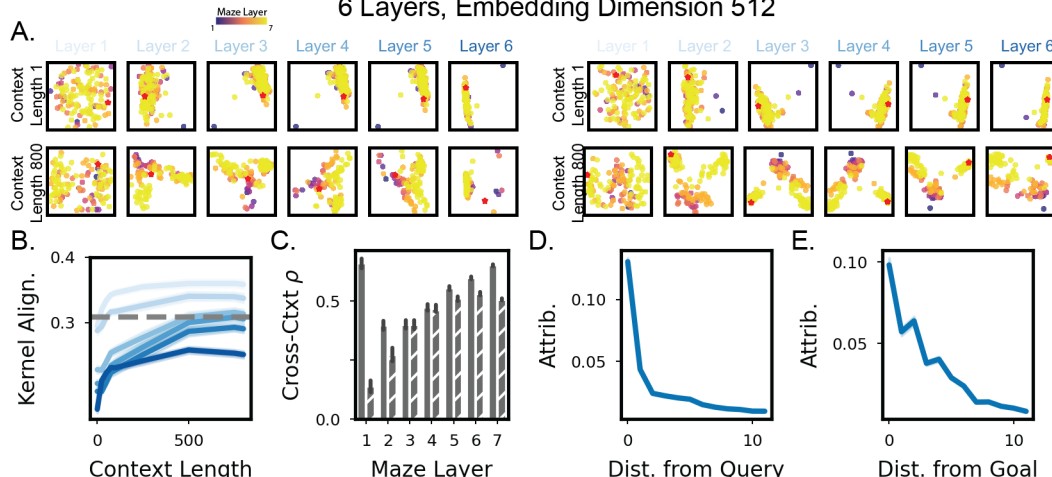

Figure 11: Sensitivity of results to model size. **A.** As in Fig 3E, but for two random example environments (left and right). Additionally, the model has twice the number of layers (6). **B.** As in Fig 3C, but for 6-layer model. **C.** As in Fig 4F, but for 6-layer model. **D., E** As in Fig 5BC, but for 6-layer model.

# E    ADDITIONAL BEHAVIORAL RESULTS IN GRIDWORLD AND TREE MAZES

Here, we show additional learning results for both the gridworld task and tree mazes. We first show additional in-context learning curves for gridworld (Fig 12). As in the main results, performance is evaluated from query states that were seen in-context and at least 6 steps away from the goal. If no eligible query states meet the selection criteria, return is recorded as zero. We note that in-context learning can still be unstable at times. In part, this may be because the model is sometimes tested on states it has not experienced. Thus, it is more difficult to navigate into previously experienced territory to find the goal. We also suspect that improvements in the training procedure or architecture that we have not explored could also produce a more performant model.

A few more behavioral results are shown for gridworld. We reproduce the analysis of Fig 2F for the meta-learned gridworld model (Fig 13A). We further subdivide the query states, however. This is because we were curious if the agent would perform differently for states seen only before any reward experience or states seen only after all reward experience. This turns out not to be the case, and the agent does equally well in both cases (Fig 13A). This information is useful for forming hypotheses of how the model solves the task. Due to its causal structure, this means that the model probably doesn't (solely) rely on a strategy where experiences of reward alter the processing of subsequent context memory tokens. Otherwise, the model should do poorly on states that were only seen before any reward experiences. We also give further details of the shortcut paths experiments (Fig 13BC).

Finally, we show additional in-context learning examples for tree mazes (Fig 14). Learning is much more stable in this environment, perhaps because there are useful heuristics the agent can use if at a novel state (transition towards parent node until arriving at a state that has already been experienced).

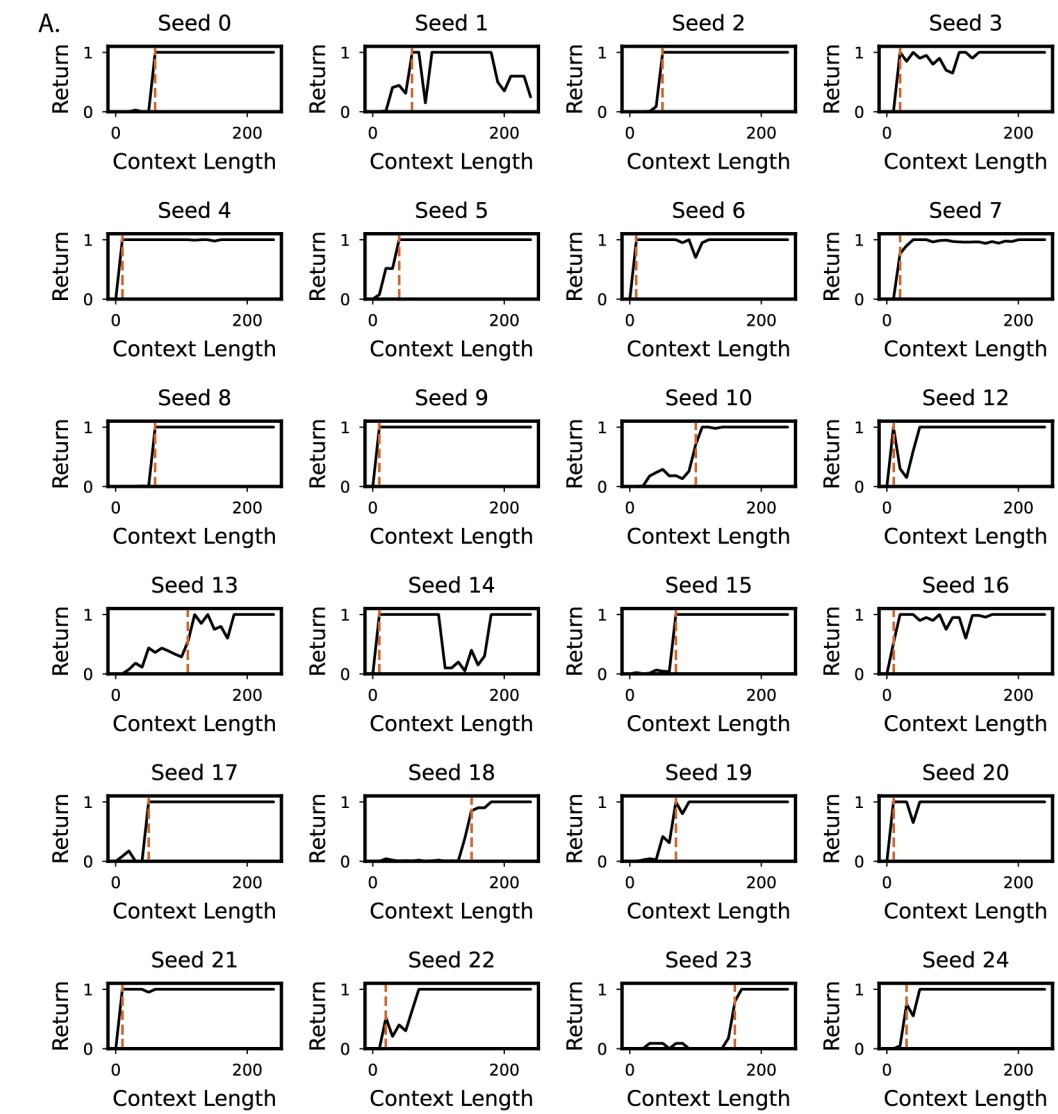

Figure 12: Additional in-context learning curves for gridworld task. **A.** As in Fig 2A, but for 24 additional test environments. We skipped environments where reward was never seen in-context.

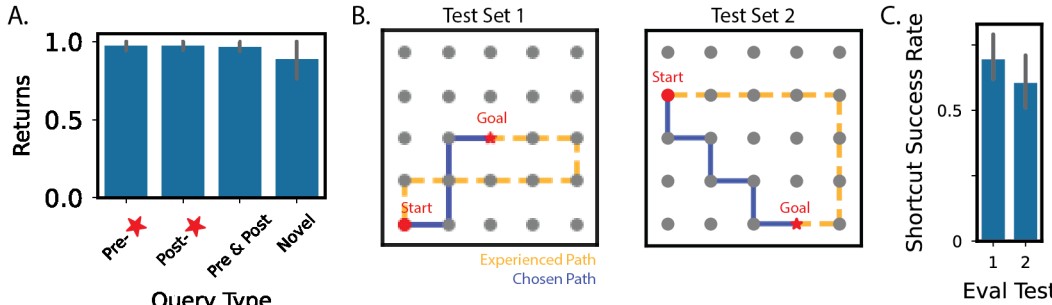

Figure 13: Additional learning results in gridworld task. **A.** As in Fig 2F, but for gridworld task. We also further divide the query states into states seen only before any reward experience (Pre-), states seen only after all reward experiences (Post-), states seen before and after reward experiences (Pre & Post), and states that were never seen in-context (novel). **B.** Depictions of the two tests for shortcut paths we use (left and right). Each test set has a fixed start location, goal location, and in-context experienced path (yellow dashed line). For each test set, we simulate 100 environments with different sensory encodings. Blue line shows an example successful shortcut path taken by the model. See methods description for more details. **C.** Success rate of taking the optimal, shortcut path for the two test sets in (B), across 100 sample environments. Error bars show $95\%$ confidence interval. Note that chance level in both tests is 0.02 (generously excluding the stay transition from consideration).

A.

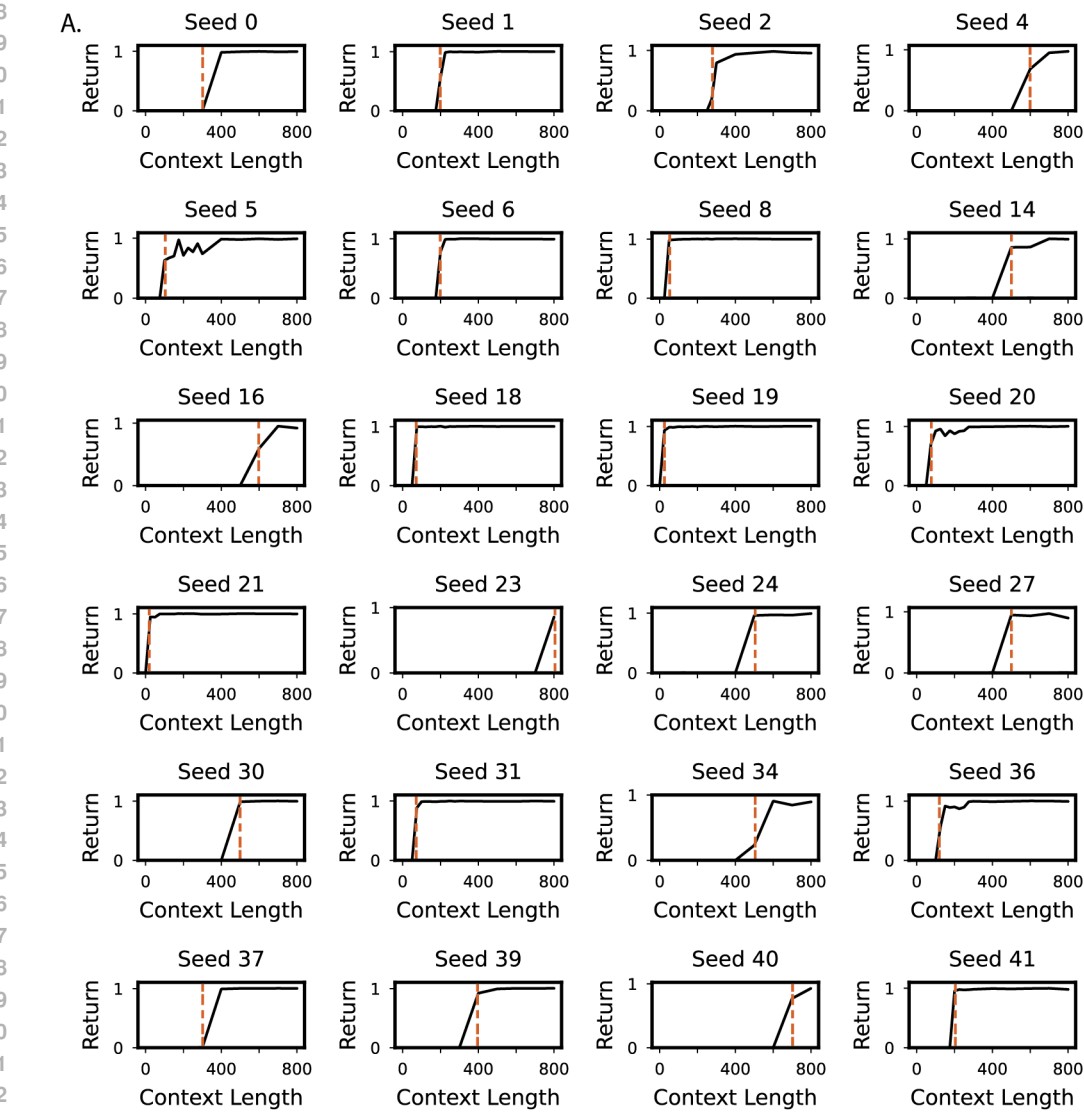

Figure 14: Additional in-context learning curves for tree maze task. **A.** As in Fig 2D, but for 24 additional test environments. We skipped environments where reward was never seen in-context.

# F Q-LEARNING SIMULATIONS

To make comparisons to RL algorithms without learned priors, we simulate two Q-learning models. We use a tabular Q-learning model where we abstract away the problem of representation learning and allow the model to use a lookup table. We also use a deep Q network (DQN) parameterized as a MLP with 4 hidden layers of dimensions $[256, 128, 64, 16]$.

To make as fair a comparison as possible, we give our Q-learners a full replay buffer and let models train to convergence on the memories of the buffer. For instance, given a task with context memory $\mathcal{D}$ if we wish to evaluate the model at context length $t_C$ we define a replay buffer comprising $\mathcal{D}_{1:t_C}$. We then let the Q-learning model train on several epochs over the full dataset of the replay buffer, until the temporal difference error has converged. We find that 1000 epochs for the tabular model and 1500 epochs for the DQN is more than sufficient to ensure this. For the tabular model, we train with batch size 512 and learning rate 0.1. For the DQN, we train with batch size 1024 and learning rate $1 \times 10^{-5}$.

There are also a few additional training details for the DQN. We randomly reinitialize the network weights at each context length before we run the training procedure. This is because we find that resetting the weights works empirically better than initializing with the weights from the previous context length the model was trained on (this is reasonable, as the latter induces a continual learning problem). To maintain as many parallels to standard methods as possible, we also adopt a double deep Q learning framework (Van Hasselt et al., 2016). We use a target network that is updated every 10 epochs. We don't think this detail is critical (and empirically the use of a target network here doesn't seem to impact performance) as the learning problem in our setting is fully stationary.

For both models, at test time we also allow for action sampling with some temperature. We empirically select the temperature that results in the best performance after a grid search over the values $[0.005, 0.01, 0.05, 0.1, 0.2, 0.5, 10.0]$. We also selected a value of $\gamma$ in the TD loss function that worked well in practice after a grid search over the values $[0.7, 0.8, 0.9]$: $\gamma = 0.8$ for the tabular model, $\gamma = 0.9$ for DQN.

# G    ADDITIONAL IN-CONTEXT REPRESENTATION LEARNING RESULTS

Here, we show additional in-context representation learning examples for more randomly sampled environments (Fig 15 for gridworld, Fig 16 for tree maze). In addition, we show the results of reward ablation on representation learning. Comparing the model with and without reward ablation, it appears that reward information sometimes results in the reward state being pushed farther away from non-rewarding states.

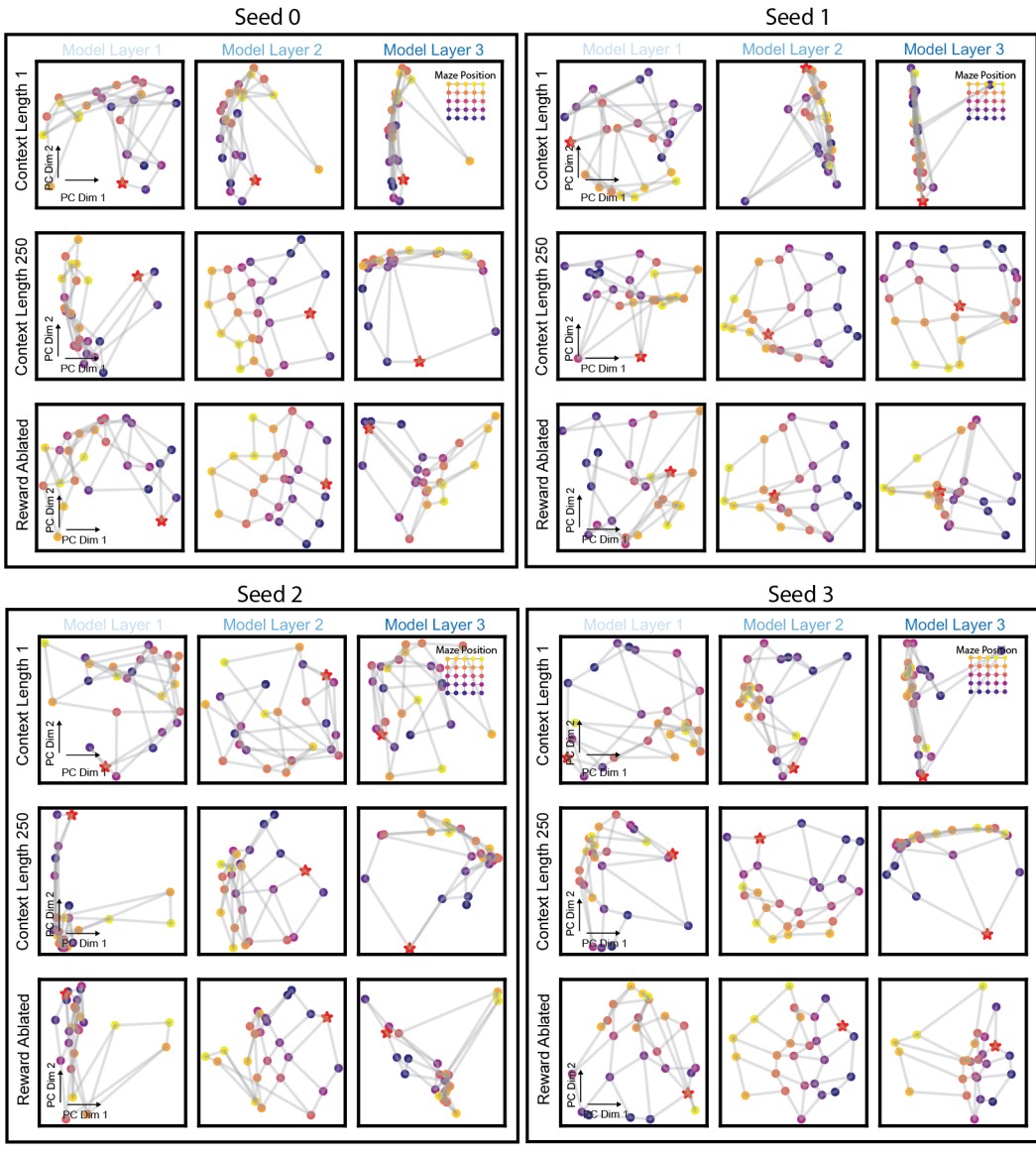

Figure 15: Additional in-context representation learning examples in gridworld task. As in Fig 3A, but for four more additional random seeds. Additionally, the third row of each plot shows the PCA embedding plots at context length 250 if reward was ablated ($r = 0$ in all transitions).

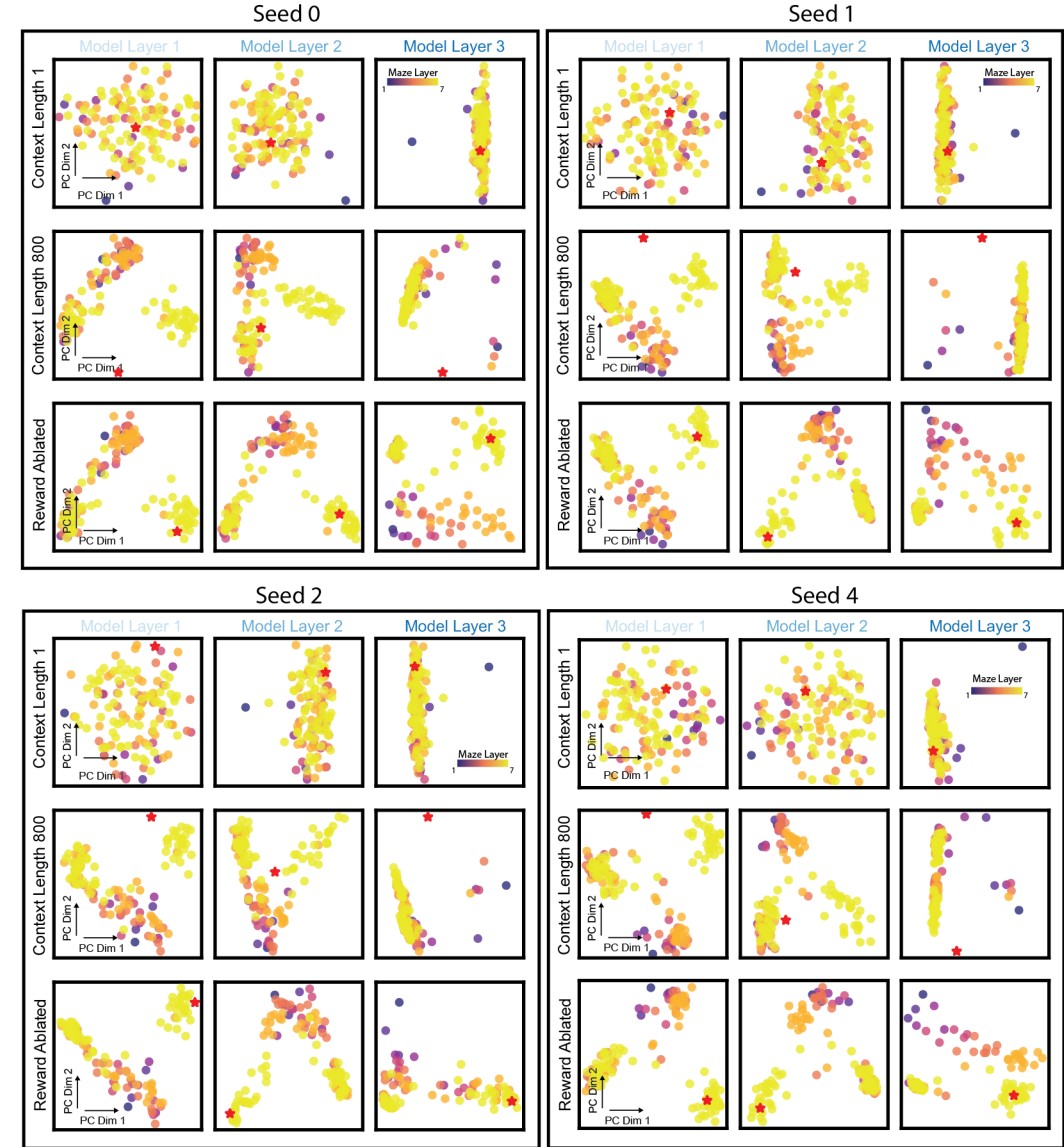

Figure 16: As in Fig 3E, but for four more additional random seeds. Additionally, the third row of each plot shows the PCA embedding plots at context length 800 if reward was ablated ($r = 0$ in all transitions). We skipped seeds where rewards was already never seen during in-context exploration.

## H  KERNEL ALIGNMENT

To quantify how well learned model representations capture the structure of the latent environment, we employed centered kernel alignment (CKA) to compare the similarity between the true environment structure and the model's internal representations. We first constructed a ground truth kernel matrix by computing the environment distance matrix $D$ where $D_{ij}$ indicates the number of actions needed to navigate from state $i$ to state $j$. We then applied an exponential transformation $K_{\text{input}} = \gamma^D$ where $D$ is the distance matrix and $\gamma$ controls the spatial scale of environment structure captured by the kernel.

For each network layer, we extracted hidden state representations corresponding to each environment state, using the final token representation as the state embedding. We collect this in the matrix $X$ and construct representation kernel $K_{\text{latents}} = (X - \bar{X})(X - \bar{X})^T$. We then compute the CKA between $K_{\text{input}}$ and $K_{\text{latents}}$.

In Fig 17AC, we show how the kernel alignment score changes for different values of $\gamma$. For the analyses in the main text, we select a value of $\gamma$ that maximizes overall kernel alignment: $0.8$ for gridworld, and $0.6$ for tree mazes.

We note that the kernel alignment measure is likely still imperfect for what we want to quantify, especially in tree mazes. For instance, the bifurcating structure of representations in tree mazes is an interesting phenomena we would like to understand better. However, it is a coarse structure that likely does not align well to the ground truth kernel that we defined. For instance, in Fig 17BD, we see how reward ablations affect kernel alignment. In the tree maze task, there appears to be a significant difference in kernel alignment that is induced by reward ablations. In contrast, the PCA plots from Fig 16 show that the branching structure of the representations is well-preserved even when rewards are ablated. Thus, we think additional metrics may be more useful to interpret representation organization in tree mazes.

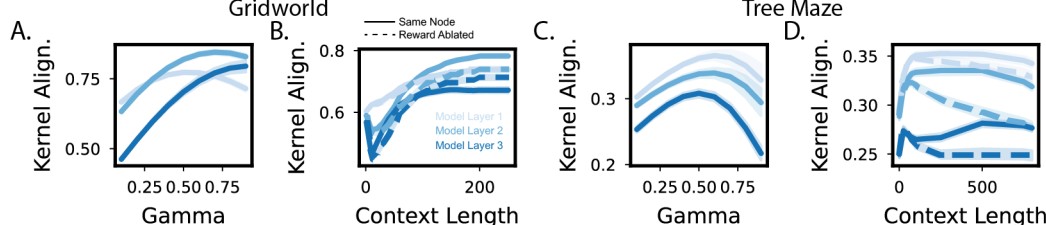

Figure 17: Additional kernel alignment details. **A.** Kernel alignment for different values of $\gamma$, the spatial kernel used to define the environment structure. Line colors indicate model layer where representations are extracted. **B.** As in Fig 3C, but showing additional lines (dashed) where reward was ablated ($r = 0$ in all transitions). **C, D** As in (A, B), but for the tree maze task.

# I    ANALYZING CROSS-CONTEXT LEARNING IN GRIDWORLD

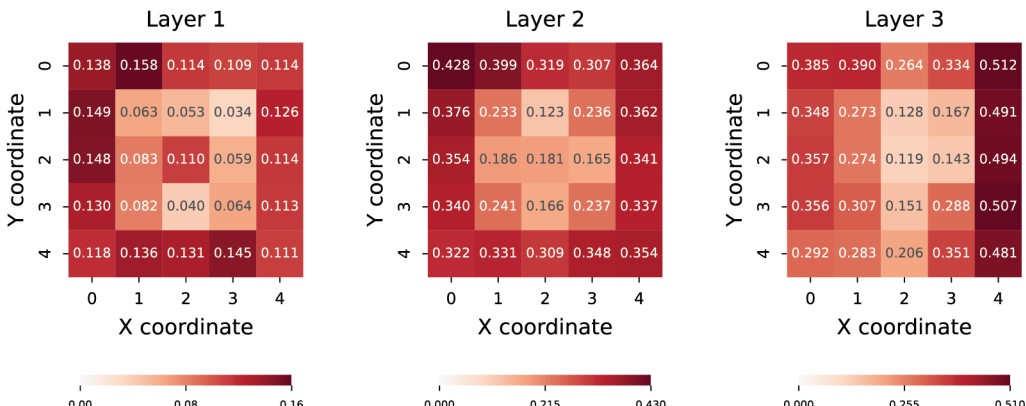

Figure 18: Additional results for cross-context representations in the gridworld task, for each of the three layers of the model. Each plot shows the difference between same-node and different-node correlations at context length 250 (that is, the difference between solid and dashed lines in Fig 4B). Values are separated by the same-node identity, i.e. the underlying XY latent state.

Here, we show additional results for cross-context representation learning in gridworld. We plot the difference between same-node and different-node correlations and separate these values by the underlying XY latent state. Potentially, representations are better aligned across contexts at the edges of the environment (see "Layer 1" and "Layer 2" of Fig 18). Overall, though, the cross-context similarity is fairly similar across the entire gridworld structure.

## J   Linear Decoder Setup

We will first describe how we linearly probe the representations of query state tokens. In gridworld, we randomly select 600 tasks from the original train set and partition these tasks into a new train/test set for our linear decoder, with a $90/10$ split. In gridworld, we use the original train set because there's more unique XY goal locations (21) than in the original eval and test sets (2 each). In tree maze, we randomly select 600 from the original test set and make the same $90/10$ train/test split for our linear decoder. We skip over tasks where reward is never seen during the in-context exploration phase.

The regressors for our decoder will be model representations at some layer. To collect them, in each environment we first identify the set of states that had been seen in-context. For each state $s'$ that was seen, we let query state $s_q = s$ and present the in-context exploration trajectory and $s_q$ as inputs to the model. For each model layer $l$, we collect the model representations for the $s_q$ token, $r(s_q, l) \in R^{512}$. The decoding task is to predict some value $v$ given $r(s_q, l)$, where $v$ is typically some kind of information pertaining to $s_q$. We tried a variety of values $v$ and in the main text only discuss the variables for which test decoding accuracy was high.

To fit a linear decoder, we use ridge regression. We standardize features to 0 mean and unit variance. The regularization strength $\alpha$ was selected through 5-fold cross-validation using a grid search over regularization strengths from $[10^0, 10^4]$, with 10 logarithmically-spaced values. Cross-validation was performed with shuffled splits. For each $\alpha$, we computed the mean $R^2$ score across all validation folds and selected the $\alpha$ that maximized this cross-validation performance. The final decoder for each layer was fit on the complete training set using the $\alpha$ found previously. Model performance was evaluated on the held-out test set.

For circular variables such as angles, we cannot directly apply standard regression since the circular nature of the data violates the assumptions of linear models (e.g., an angle of $\pi$ and $-\pi$ represent the same direction but appear numerically distant). Instead, we decompose each target angle $\theta$ into its sine and cosine components: $\sin(\theta)$ and $\cos(\theta)$. We then fit two separate ridge regressors to predict these components independently, using the same cross-validation procedure described above. To obtain the final angle prediction, we convert the predicted sine and cosine values back to angles using the arctangent function: $\hat{\theta} = \arctan2(\sin(\hat{\theta}), \cos(\hat{\theta}))$.

For classification tasks, we do the same but with logistic regression and report balanced accuracy scores.

To probe representations for context memory tokens, we follow a similar procedure as that for query state tokens. In each environment, we pass the entire in-context dataset $\mathcal{D}$ to the model. We then iterate through $t = [T, T-2, \ldots, 1]$ and collect representations from the model in response to token $\mathcal{D}_t = (s_t, a_t, s'_t, r)$ if the transition $(s_t, a_t, s'_t, r)$ has not already been collected for this environment. We work backwards under the assumption that model representations are more rich as in-context experience increases, and thus more likely to contain task-relevant variables. For each token that produces a regressor, we define variables of interest relative to $s_t$ (e.g., value function for state $s_t$). We did not see a difference when we defined the variables we tested relative to $s'_t$ instead.

## K   Gradient Attribution Method

To get gradient attributions, we use integrated gradients (Sundararajan et al., 2017). As a reminder, the model output is a vector defining weights over actions. We calculate the gradient of the model's output for the optimal action with respect to input tokens. We define the baseline inputs as the original context memory dataset $\mathcal{D}$ but with actions ablated (that is, $a = \mathbf{0}$). We integrate over 20 steps.

## L  TESTS FOR MODEL-FREE REASONING

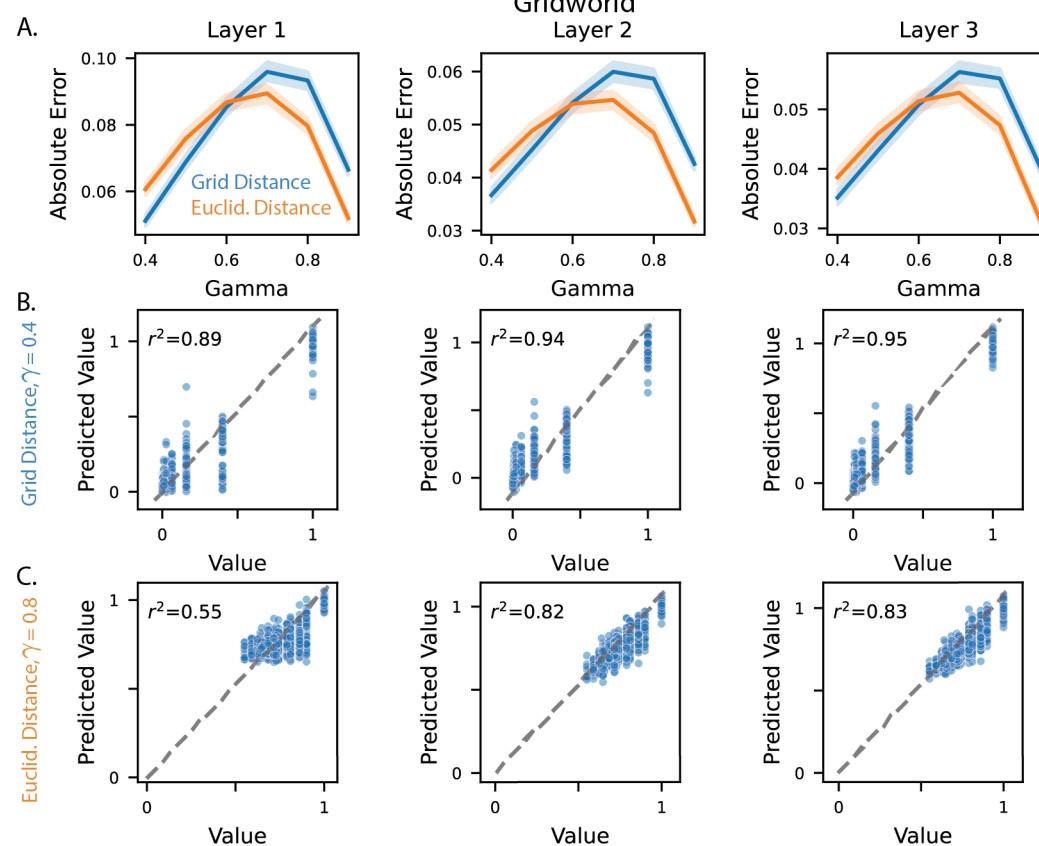

Figure 19: Tests for model-free reasoning in gridworld. **A.** Absolute error in test set for decoders fit on $V^*$ (Grid Distance, blue) and $V_e$ (Euclid. Distance, orange) across different values of $\gamma$. **B.** Predicted value vs actual value in test set, for $V^*$ and $\gamma = 0.4$. **B.** Predicted value vs actual value in test set, for $V_e$ and $\gamma = 0.8$.

As a probe for model-free reasoning, we tested whether or not value information could be decoded from model representations. Specifically, we test whether, at decision time, the model utilizes value information of the query state to drive decisions.

Specifically, we assess whether the model encodes value estimates $V^*(s) = \mathbb{E}_\pi\left[\sum_{t=0}^\infty \gamma^t R_{t+1} \mid S_0 = s\right]$ for its current state $s$ under an optimal policy. We trained linear decoders on query token representations to predict $V(s)$, but overall did not find evidence that a value gradient could be extracted from model representations (see Apps. K and L). In gridworld, $V(s)$ can be decoded with high accuracy (Appendix L). However, decoding is more accurate when $V(s)$ is defined in terms of Euclidean distance to the goal, rather than over the true 4-dimensional action space (Appendix L). This suggests that the model encodes spatial structure rather than true value gradients—its apparent $V(s)$ reflects geometric regularities, not action-contingent reward prediction. In tree mazes, decoded value estimates are localized: $V(s)$ is only reliable within 2–3 steps of the reward (Appendix L). This narrow value gradient is insufficient to guide behavior over the full task horizon.

We start with fitting linear decoders in gridworld. Let $s$ be a query state and $s_{goal}$ be the reward state. The variable we predict from the model representations is $V^*(s) = \mathbb{E}_\pi\left[\sum_{t=0}^\infty \gamma^t R_{t+1} \mid S_0 = s\right]$ for state $s$, taking an optimal policy. Equivalently, $V^*(s) = \gamma^{d(s,s_{goal})}$, where $d(s, s')$ describes the number of actions needed to navigate from $s$ to $s'$. Thus, $V^*$ describes an exponentially decaying value gradient in terms of action distance. To evaluate the decoder, we plot the test error against

the value function $\gamma$ (Fig 19A). We note that, although the lowest error is achieved at $\gamma = 0.4$ (Fig 19B), this is not actually a useful parameterization for a value function as the value gradient decays quickly for states more than 2 steps away from reward. However, the test error at $\gamma = 0.8$ is as low as $0.04$ in the final model layer.

Given that model representations capture the environment structure well, we suspect that the high decoding accuracy for $V^*$ may result from the spatial organization of representations. That is, if XY location information is contained in representations, a linear decoder could also do fairly well at predicting $V^*$. As a control, we define $V_e = \gamma^{d_e(s,s_{goal})}$ where $d_e(s,s')$ describes the Euclidean distance from $s$ to $s'$. We find decoding error is lower for $V_e$ than for $V^*$. However, $V_e$ does not reflect the actual action affordances in gridworld (since action space is only up/right/left/down). Thus, we conclude that the strategy used by the model may have more to do with learning the latent Euclidean structure of the environment than learning a value function across action space (as would be expected in standard model-free algorithms).

We repeat this analysis in the tree maze task. We define $V^*$ as before and plot the test error against the value function $\gamma$ (Fig 20A). We find that decoding error increases with $\gamma$. We plot predicted $V^*$ vs actual $V^*$ for the lowest and highest $\gamma$ values (Fig 20BC). We find that at $\gamma = 0.4$, $V^*$ is well fit, however the value gradient is only meaningful for states that are 1-2 steps away from reward. Thus, this is likely not useful as a model-free RL signal. Conversely, at $\gamma = 0.8$, the decoding does not perform well, and at most reaches $r^2 = 0.53$.

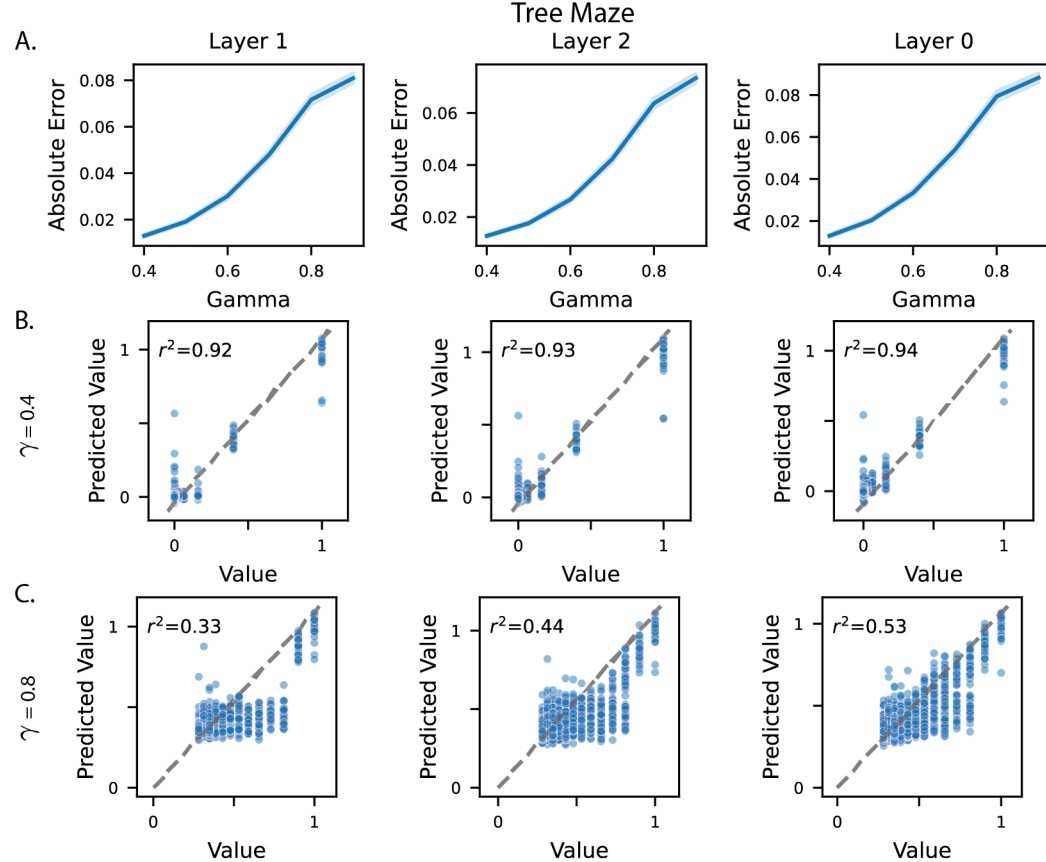

Figure 20: Tests for model-free reasoning in tree mazes. **A.** Absolute error in test set for decoders fit on $V^*$ across different values of $\gamma$. **B.** Predicted value vs actual value in test set, for $V^*$ and $\gamma = 0.4$. **B.** Predicted value vs actual value in test set, for $V^*$ and $\gamma = 0.8$.

# M Tests for Model-Based Reasoning

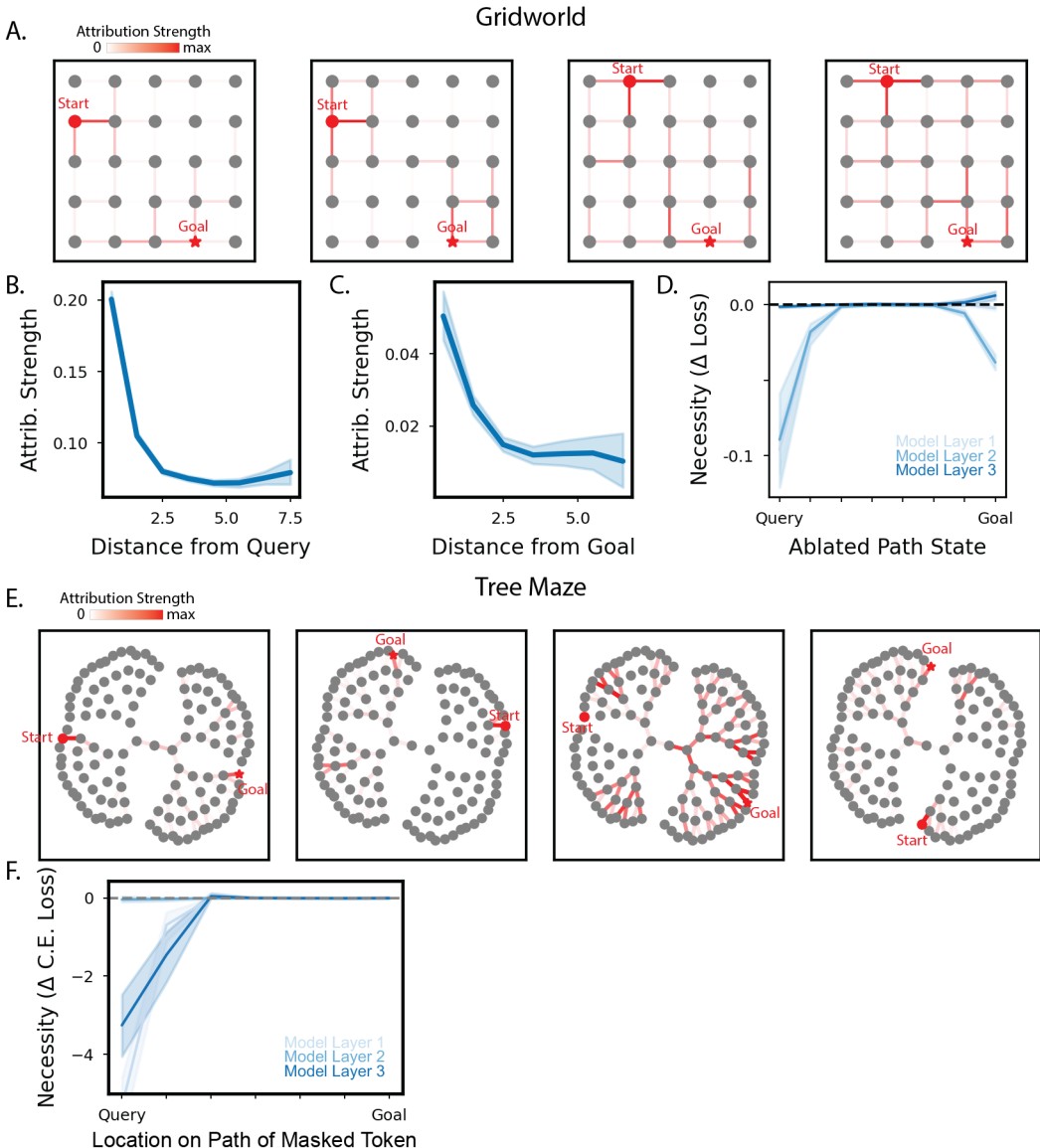

Figure 21: Tests of model-based reasoning. **A.** As in Fig 5A, but for four example gridworld environments. **B., C.** As in Fig 5BC, but for gridworld tasks. **D.** Measurement of necessity for each context memory token on the path from the query state to the goal state. We measure the change in cross–entropy loss when the token is ablated, and we plot this against the location of the token. Tokens are ablated by masking the query-to-token attention at the desired model layer. In all tests, the query state remains the same. Line color indicates which layer of the model the intervention was conducted in. We show average value across 50 environments, with 95% confidence interval shading. Since there are multiple possible paths from query token to goal in gridworld, we define the path as the sequence of states the agent would have taken had we allowed it to navigate to reward. We include only cases where the model successfully navigates to goal. **E.** As in Fig 5BC, but for four more tree maze examples. **F.** As in (D) but for the tree maze task.

We next probe for signatures of model-based reasoning. That is, we look for evidence that the model utilizes path planning to choose the correct action from the query state. This is connected to questions of state tracking (Li et al., 2025) and understanding how models simulate successive transitions between states. Li et al. (2025) propose different ways that transformer models can do this

path planning, from forward rollouts to more sophisticated, mergesort-like algorithms. Each of these algorithms require simulating transitions through intermediate states between query and goal. Thus, to test for the presence of path planning, we look for evidence that information about intermediate states are utilized at decision time. Specifically, we isolate decision time as computations conducted in the query token stream.

We first begin with gridworld and analyze the gradient attributions over the input context tokens (which are themselves transitions). We plot individual examples of these attribution maps and summary statistics in Fig 21A-C. Taken together, it does not appear that the model relies on path planning from the current state to the goal. We further test with ablations of states on the path from query to goal. Specifically, at each model layer we conduct a necessity test where we mask attention from the query token to context tokens containing the ablated state. We then measure how attention ablations impact the original cross-entropy loss (Fig 21D). We find that ablating intermediate states does not impact cross-entropy loss.

We conduct the same analyses in tree maze (Fig 21EF) and find similar results. We conclude that path planning as done in typical model-based reasoning is not a strategy that the model is relying on to solve either tasks.

Our conclusion contrasts with prior work that analyzed the behavior of meta-learned RL agents and suggested that they implement a form of model-based planning (Wang et al., 2016; Ritter et al., 2018). It is possible that meta-learning can discover more typical model-based strategies in other settings, and these discrepancies demonstrate that the mechanistic perspective taken here can provide insights into understanding meta-learned algorithms.

# N    GRIDWORLD MECHANISMS

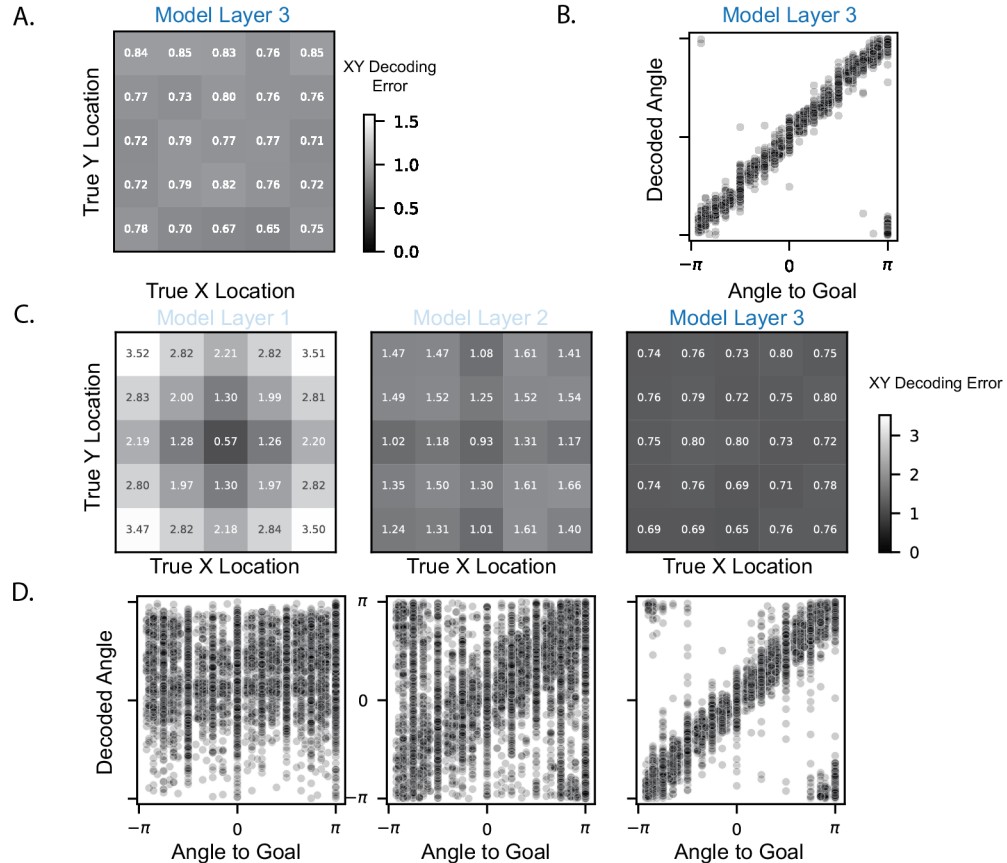

Figure 22: Additional decoding plots for gridworld task. **A., B.** As in Fig 6BC, but for model layer 3. **C.** As in Fig 6B, but decoding analysis is run on the context memory tokens that enter each layer of the model. **D.** As in Fig 6C, but decoding analysis is run on the context memory tokens that enter each layer of the model.

In Figure 22, we show additional decoding results for gridworld. In the main text, we discussed how XY location and angle to goal can be decoded clearly from the query token stream of the model. We also find that these variables can be decoded from the context tokens. We find this interesting as it connects to our findings in tree mazes where memory tokens contain not just information about the original event (i.e., transition), but also additionally computed features.

In results section 3.5-Gridworld, we gave brief descriptions of a few analyses we ran. Here we will give more details on these analysis. First, we discussed how we showed that model performance relies on attending to tokens near the query and goal states in layer 2 (Fig. 6D). We do this by masking attention from the query token to individual context tokens, following the ablation procedure described in Section 3.4. We find that layers 1 and 3 are robust to these ablations, but performance degrades in layer 2 when attention to tokens near the query or goal is removed (Fig. 6D).

We also discussed how the attention patterns between context memory tokens shift from localized to distributed across model layers (Fig. 6), suggesting that the model first stitches transitions locally before constructing global structure. To arrive at this conclusion, we analyze the spatial locality of attention between context-memory tokens to test whether transitions are integrated locally or globally. Specifically, we plot attention strength as a function of spatial distance between token pairs (Fig. 6E). We restrict this analysis to the first two layers, since context-to-context attention in the final layer does not influence the policy output.

## O  TREE MAZE MECHANISMS

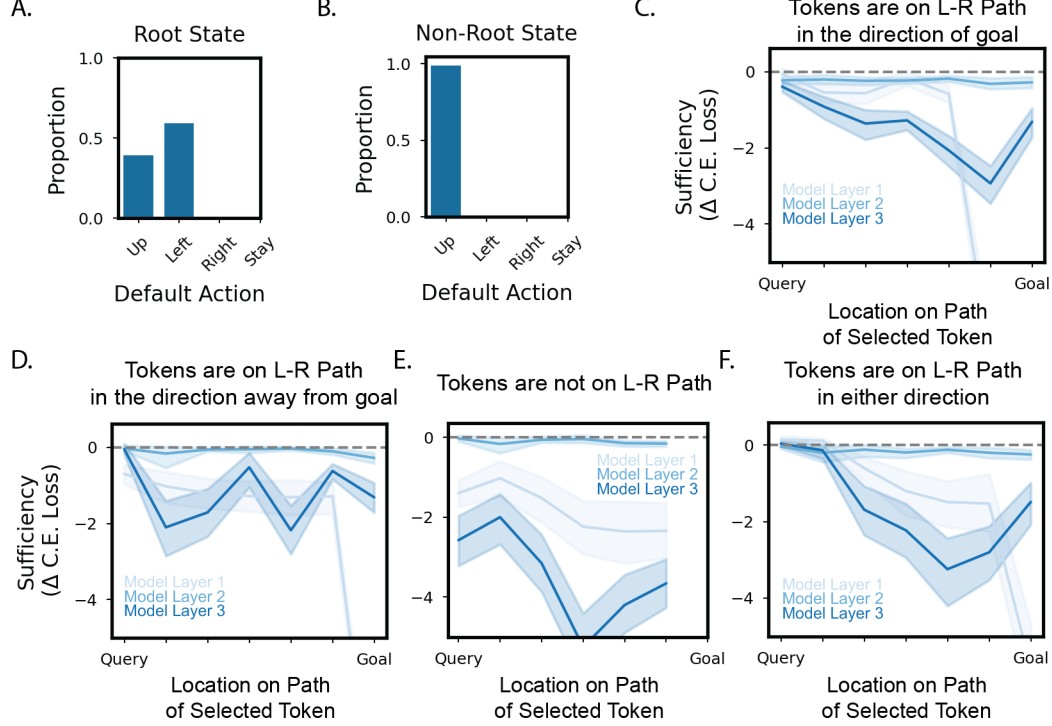

Figure 23: Additional analyses for tree maze task. **A.** Across 50 environments where reward information is ablated from the context, the proportion of each action taken by the model when at the root state. **B.** As in (A), but for non-root states. **C.** As in Fig 7C, but we further restrict the context memory tokens to be on the L-R path and transitioning in the direction towards goal. **D.** As in (C), but transitioning in the direction away from goal. **E.** As in (C), but for context memory tokens that are non the L-R Path. That is, for the state indicated on the x-axis, we select context memory tokens that involve that state but are not on the L-R path. There are no transitions involving the goal that are not on the L-R path, and thus no data at that point. **F.** As in (C), but we restrict context memory tokens to be on the L-R path and do not further restrict by their directionality to or from the goal.

In Figure 23, we show additional results for our tree maze analysis. In the tree maze task, the optimal action is often to transition to the parent node (specifically, this is true in all but the 6 states that comprise the L-R path). This bias is reflected in the model. Without reward information, the model defaults to transition towards the parent node unless it is at the root node (Fig 23AB). Thus, we believe the model takes its default action unless it accumulates enough evidence through its layer computations to do otherwise. As discussed in the main text, we think this is done by tagging context tokens on the L-R path.

We also give more information here about analyses that we briefly described in results section 3.5-Tree Maze. We discussed a hypothesis where, at decision time, the model tests if there are context-memory tokens that contain the query state and are tagged as being on the L-R path. If so, then the correct left/right action can be inferred from the same tagged tokens (in particular since inverse actions are also encoded).

We find further evidence for this strategy by re-doing our sufficiency analysis from Fig 7C with three additional restrictions: (1) tokens must be on the L-R path in the direction to the goal, (2) tokens must be on the L-R path in the direction away from the goal, or (3) tokens are not on the L-R path at all. We find that in the first two cases the model output is unaffected, but in the third case the output is negatively impacted ((Fig 23C-F)). As long as the query token in the last layer can attend to context tokens that (1) themselves contain the query token and (2) are on the L-R path going towards or away from goal, the model output is unaffected (Fig 23C-F). The results of these perturbations are

consistent with our hypothesis that decision-making in the model relies on identifying if the query state is on the L-R path via intermediate computations stored in context-memory tokens.