# OpenReview forum: "Memories to Maps: Mechanisms of In-Context Reinforcement Learning in Transformers"
_ICLR.cc/2026/Conference — Submitted to ICLR 2026_

### Official Review · Reviewer_5CU8 · 2025-10-23

**Soundness:** 2
**Presentation:** 3
**Contribution:** 3
**Rating:** 4
**Confidence:** 5

**Summary:**

This paper investigates how transformers learn to perform in-context reinforcement learning in navigation tasks inspired by rodent behavior. The authors train models on gridworld and tree maze tasks, then analyze the representations and strategies that emerge. They claim the model develops structure learning capabilities and memory-based computation strategies that resemble hippocampal-entorhinal computations, while being neither purely model-free nor model-based RL.

**Strengths:**

1. Interesting research question: The investigation of how memory-based computation supports rapid adaptation is timely and relevant for understanding both artificial and biological intelligence.
2. Dual task design: Using both gridworlds and tree mazes provides useful contrasts between Euclidean and hierarchical structure.
3. Comprehensive analysis: The paper includes representation analysis, attention patterns, and decoding studies across multiple dimensions.
4. Clear visualizations: Figures effectively communicate the key phenomena (e.g., Figs 3-4 showing representation organization).

**Weaknesses:**

## Major Issues

### Overstated Neuroscience Claims
Problem: The abstract and introduction make strong claims about resemblance to hippocampal-entorhinal (HC-EC) computations, but the evidence is thin and the connections are loose analogies rather than mechanistic correspondences.

* The claim that "representations developed in the model resemble computations associated with the hippocampal-entorhinal system" (abstract) is not well-supported. The authors show structure learning and cross-context alignment, but these are general computational principles, not specific to HC-EC.
* Section 3.3 cites Whittington et al. (2020) as the primary reference, but this represents one theoretical perspective, not consensus. The authors should cite broader literature (e.g., Buszaki, O'Keefe, Moser work on place cells, grid cells, etc.) and be more measured in claiming parallels.
* The Stachenfeld et al. (2017) citation (line 269) regarding "predictive map * * formation in the hippocampus / long hypothesized as a computational scaffold * for memory" overstates the acceptance of this particular theory. While certainly interesting, successor representations are one of several competing theories.

Recommendation: Substantially tone down neuroscience claims or provide much more
detailed mechanistic comparisons. The work stands on its own as an AI
contribution without requiring strong brain analogies.

### "Computational Substrate" Framing (Introduction, paragraph 2)
Problem: The opening sentence "Here, we ask how episodic memory operates not
just as storage, but as a computational substrate for rapid learning and
decision-making" is conceptually muddled.

* What precisely is meant by "computational substrate"? This seems to conflate memory-as-storage with memory-as-computation in a way that's not clearly defined.
* The distinction being drawn is unclear: obviously neural networks don't operate like von Neumann architectures, so what new insight is being claimed?

Recommendation: Clarify the specific computational hypothesis. Are you asking
whether memory stores raw experiences vs. computed features? Whether retrieval
is passive vs. active computation? State this precisely.

### Euclidean vs. Manhattan Distance
Problem: Line 82 claims gridworlds are "Euclidean and spatially continuous" but
standard gridworlds use Manhattan (L1) distance due to movement constraints
(up/down/left/right).

Evidence: The shortest path metrics and action space (4-directional movement)
suggest Manhattan geometry, not Euclidean (L2).

Recommendation: Clarify the geometry. If you mean the learned representations
embed into Euclidean space (Fig 3B), state this clearly rather than claiming the
environment itself is Euclidean. The distinction matters for interpreting
shortcut behavior.

### Context Length Confound (Section 3.1, Fig 2)
Critical Problem: The rapid learning demonstrated in Fig 2 may simply reflect
memorization of the entire environment rather than genuine one-shot learning.

* As context length increases, the transformer has access to more state transitions, potentially covering the full state space.
* No analysis is provided of: (a) what percentage of the state space is covered at each context length, (b) whether query states are actually novel or have been seen in context.
* The comparison to Q-learning is confounded because Q-learning must learn representations while the transformer may simply memorize.

Recommendation:
1. Report state space coverage as a function of context length
2. Separately analyze performance on truly novel states vs. states seen in context
3. Without this analysis, claims about "one-shot learning" (line 198) are not supported

### "Inverse Actions" Claim (Section 3.1, lines 205-210)
Problem: The authors claim the model "acquires a useful prior: the ability to infer inverse actions" based on better performance on post-reward query states compared to tabular Q-learning.

Issue: This doesn't demonstrate inverse action inference. It could reflect many other factors:
* Better generalization from meta-learning
* Value propagation through attention mechanisms
* Implicit model building

Recommendation: To claim inverse action learning, directly test whether the model can predict inverse actions or show that interventions on representations related to inverse actions affect this specific performance gap.

### Structure Learning Claims (Section 3.2)
Problem: Lines 212-213 ask "does a structured representation learning strategy emerge during in-context processing?" but the answer is ambiguous.  Issues:

* The task structure (gridworld topology, reward-driven behavior) is implicit in the training objective, so representations aligned with task structure shouldn't be surprising
* The authors claim "no such objective was imposed during training" (line 230), but this isn't entirely truthful. The supervised training on optimal actions implicitly requires understanding structure, and requires clarification what the authors mean.
* Calling this "spontaneous" structure learning (implied throughout) overstates the result

Recommendation: Acknowledge that structure learning is implicitly encouraged by the training objective. The interesting finding is how the model represents structure, not that it does.

### Model-Free vs. Model-Based Analysis (Section 3.4-3.5)
Major Conceptual Problem: The authors claim their model is "neither model-free nor model-based" but the evidence is insufficient and the argument appears self-contradictory.
Issues:

Section 3.4 shows the model doesn't use explicit path planning or simple value functions
However, Section 3.5 (especially tree mazes) shows the model learns to:

* Stitch transitions backwards from goal to root
* Tag memory tokens on critical paths
* Store inverse actions and path membership

This sounds very much like model-based planning with a compressed/abstract model, and not "outside the standard taxonomy."
Alternative interpretation: The model learns a compact, abstract world model stored distributedly across memory tokens, then uses this for planning. This would be compressed model-based RL, not a new category.

Recommendation:
* Test whether the model learns intermediate state-to-state connectivity (even without full explicit path enumeration)
* Consider whether "cached computations" are actually compressed model features
* If the model truly is neither, provide a formal characterization of what category it does fall into

### Tree Maze Analysis (Section 3.5)
Problem: The mechanistic description for tree mazes seems to contradict the earlier "not model-based" claim.

The proposed strategy (lines 418-422):
* Stitch transitions backwards from goal to root
* Tag memory tokens on L-R path
* Check if query state is on L-R path
* Extract optimal action if yes, default to parent otherwise

This is path planning with a learned model, exactly what model-based RL does, just implemented through memory tagging rather than explicit rollout.
Recommendation: Reconsider the categorization or explain precisely why this doesn't constitute model-based planning.


## Minor Issues

1. "Task distribution" terminology (line 143): For gridworlds with fixed structure and varying rewards, "task distribution" is confusing. Perhaps "task family" or "task ensemble" is clearer.
2. Unmotivated claim (line 158-159): "(which may be a relevant analogy for language generation tasks)" needs citation or elaboration. What specific aspect of language generation maps to tree mazes?
3. Methods brevity (Section 2): Critical details are relegated to appendices, making reproduction difficult. At minimum, provide:
  * Full architecture specifications in main text
  * Training hyperparameters
  * State encoding details
  * Exploration policy details
4. "Mechanistic analysis" (contribution list, line 87): What precisely is meant by this? The paper contains representational analysis and intervention studies, but "mechanistic" suggests a level of understanding of cause-effect relationships that isn't fully achieved.
5. Transformer as episodic memory (lines 108-110): The claim that key-value architectures "resemble episodic memory systems in the brain" cites only Krotov & Hopfield (2020) and related work. This perspective is not universally accepted, please cite contrasting views or soften the claim.
6. Statistical reporting: Many figures show 95% confidence intervals but don't report:
  * Sample sizes for all analyses
  * Multiple comparison corrections where applicable
  * Effect sizes
7. Reward ablation (Fig 3D, 3H): The finding that structure learning is "largely unaffected by the presence of reward" (line 255) needs more discussion. This seems surprising given that reward location should be crucial for goal-directed behavior.
8. Correlation vs. causation (throughout Results): The authors frequently imply causal relationships (e.g., "representation learning strategies support in-context RL") but only show correlations. Intervention studies or ablations are needed for causal claims.
9. Shortcut behavior (Section 3.1): The 60% shortcut rate (App E) seems relatively low if the model truly learns Euclidean structure. What explains the 40% of cases where shortcuts aren't taken?

## Missing Elements

1. LLM Use Statement: Per ICLR guidelines, authors must disclose any use of large language models in preparing the manuscript. This statement is absent.
2. Baselines: Beyond tabular Q-learning and DQN, comparisons to other meta-RL methods (RL^2, MAML, etc.) would strengthen claims about the uniqueness of the learned strategy.
3. Ablation studies:
  * What happens with different architecture choices (deeper, shallower, different attention mechanisms)?
  * How sensitive are results to training distribution characteristics?
4. Generalization tests: Do strategies learned on gridworlds transfer to tree mazes or vice versa? This would test the abstractness of learned priors.
5. Computational cost: No discussion of training time, computational requirements, or scalability.

## Detailed Comments on Specific Sections
### Abstract
* "mechanisms of in-context reinforcement learning" - too vague, specify what mechanisms
* "not interpretable as standard model-free or model-based planning" - this is a negative claim that needs stronger support

### Section 3.2
* Line 230: "crucially, no such objective was imposed during training" - misleading as discussed above
* Fig 3C: Why does layer 2 show strongest alignment? Interpretation needed.
* Tree maze representations (line 258): "coarse, high-level structure rather than fine-grained spatial layout" - how do you distinguish these? Need quantification.

### Section 3.3
* Line 306: "Do similar cross-context alignment strategies emerge in meta-learned agents?" - the answer would be more convincing with comparisons to models trained without meta-learning objective
* Missing citations to broader literature on compositionality, transfer learning, etc.

### Section 3.4
* Line 339: "limited expansion from the query state and the goal state" (Fig 5) - but expansion is visible. "Limited" needs quantification against a baseline expectation.
* No value decoding evidence is mentioned in main text (only in appendices) - if this negative result is important for your claims, include it prominently.

### Section 3.5
* The decoding analyses (Fig 6B, C) are interesting but: why these specific variables (XY position, angle-to-goal)? What about other potentially relevant variables?
* Line 377: "both XY position and angle-to-goal can be decoded from the embeddings of the context-memory transitions" - implications unclear, needs interpretation

### Conclusion

* Line 478-485: Dramatically overstates findings
* "rapid adaptation of agents in tasks relevant to natural cognition" - your tasks are simplified abstractions
* "novel use of episodic memory" - the cited Dasgupta & Gershman (2021) already proposed this, so not novel
* Missing: limitations, future work, clear takeaways

**Questions:**

1. Can you provide a formal definition of what computational framework your model does use, if not model-free or model-based RL?
2. Have you tested whether representations remain structured when controlling for state space coverage?
3. What specific hippocampal-entorhinal computational motifs (beyond general structure learning) are present in your model?
4. How does performance scale with environment size and complexity?
5. Can the learned strategies transfer across task types (gridworld -> tree maze)?

---

### Official Review · Reviewer_r84f · 2025-11-01

**Soundness:** 3
**Presentation:** 4
**Contribution:** 2
**Rating:** 4
**Confidence:** 3

**Summary:**

A key feature of the transformer architecture is attention, which is leveraged for its in-context learning capabilities. This paper investigates the algorithmic mechanisms of transformer models trained with in-context reinforcement learning (ICRL) and draws parallels to episodic memory in humans and animals, which is used for fast adaptation. Specifically, by training transformers on two neuroscience-inspired tasks using ICRL, the paper examines what representations the model learned and hypothesizes what strategies it employed on a computational level.

**Strengths:**

The paper provides an in-depth analysis of the representation and strategies learned by ICRL. The analysis looks solid and aligns with neuroscience practices.

**Weaknesses:**

The paper emphasizes that the strategies developed by ICRL cannot be interpreted as model-free or model-based planning. The arguments are weak in my view. Both model-free and model-based umbrellas are huge. Model-free method doesn't necessarily involve value function; an agent that learns a  geometric map and calculates an angle to the goal (the grid-world strategy) probably implicitly uses a "model."

While the paper adopts a meta-learning framework, it does not sufficiently situate its findings within the broader meta-reinforcement learning literature. Are we expecting the findings unique to ICRL or universally applicable to other meta-learning RL methods, in particular those with a memory component?

**Questions:**

1. Could the author comment on why the experimental setup makes sense, e.g. using random exploration transitions as contexts and the prediction of optimal policy? Are these assumptions inspired by existing neuroscience literature?

2. To what degree the conclusions and findings depend on the exact training setup such as those mentioned in the previous question and the network configuration?

3. Could the author provide some discussion about whether other meta-learning algorithm, in particular those equipped with a memory component, would exhibit the same characteristics that the paper uncovered?

4. Could the author elaborate why is the ability to predict value function is equivalent to "hallmarks of standard model-free reinforcement learning"? In particular, would that apply to policy gradient methods as well.

5. Is there a way to get more evidences supporting the proposed strategies? The decoding exercise only confirms the XY location and angle to goal information present in the representation, but doesn't show it's used for decision making. The intervention on the context only proves that the network needs to aggregate information from the query and goal states, but doesn't show the nature of the aggregated information. Overall, the proposed strategies are more like plausible hypothesis rather than proved theory.

6. Are there aspects that ICRL is dissimilar to the hippocampal-entorhinal system in the brain? Could the authors provide a comparison of ICRL to other computational models of the hippocampal-entorhinal system?

---

### Official Review · Reviewer_DtuL · 2025-11-03

**Soundness:** 1
**Presentation:** 3
**Contribution:** 1
**Rating:** 2
**Confidence:** 4

**Summary:**

This paper trained a transformer-based architecture on two navigation tasks (grid-world and tree-structured) and observed phenomena similar to those observed in biological agents. The authors concluded that their results provide a potential alternative to both model-based and model-free reinforcement Learning (RL) frameworks.

**Strengths:**

The paper is generally clear and well-written. It also tackles a significant problem: understanding the mechanisms behind in-context learning.

**Weaknesses:**

Most importantly, the presented evidence for the claim is minimal. Moreover, there is not much diversity in the tasks. The problem sets are small (e.g., 5*5 grid-word) for a deep architecture, and there are not many alternative methods for comparison. Also, since the paper used an already developed model, there is no contribution to model development.

**Questions:**

- Deep architectures are pretty powerful tools to learn patterns in general. How does a simple fully connected network or a CNN architecture perform in the tested problems?
- How do the deep model-based architectures work in these problems? For example, value iteration networks that are based on classic model-based RL are good methods to be tested. (See "Value Iteration Networks" by Tamar et al and "Generalized Value Iteration Networks:
Life Beyond Lattices" by Niu et al for reference. These works also show examples of fully connected networks and CNNs on their problem sets.
- How does the transformer-based architecture scale? Do they preserve the observed phenomena on larger sets?
- "Context" is a vague term, especially across different disciplines (here neuroscience v.s. machine learning/AI). It would be beneficial if the authors explain context and in-context learning, in a unified way in the paper.
- It is essential to test the claims in a task other than navigation to see the generalizability of the approach.

---

### Meta-Review · Area_Chair_tAfS · 2026-01-04

**Summary:**

I will list the most important comments that the reviewers noted during the review process:
1) The presented evidence for the claim is minimal and there is not much diversity in the tasks
2) The problem sets are small  for a deep architecture
3) The paper does not sufficiently situate its findings within the broader meta-reinforcement learning literature
4) The abstract and introduction make strong claims about resemblance to hippocampal-entorhinal computations, but the evidence is thin and the connections are loose analogies.
5) The rapid learning demonstrated in Fig 2 may simply reflect memorization of the entire environment rather than genuine one-shot learning.
6) The authors claim their model is "neither model-free nor model-based" but the evidence is insufficient and the argument appears self-contradictory.

**Reviewer Concerns:**

The authors did not provide answers to the reviewers' comments, so all the shortcomings noted remained in force.

**Reviewer Scores:**

1) Reviewer DtuL (score 2) would have left his initial score.
2) Reviewer r84f (score 4) would have left his initial score.
3) Reviewer 5CU8 (score 4) would have left his initial score.

---

### Decision · Program_Chairs · 2026-01-26

Reject